# Masked Prediction: A Parameter Identifiability View

**Bingbin Liu**
Carnegie Mellon University
`bingbinl@cs.cmu.edu`

**Daniel Hsu**
Columbia University
`djhsu@cs.columbia.edu`

**Pradeep Ravikumar**
Carnegie Mellon University
`pradeepr@cs.cmu.edu`

**Andrej Risteski**
Carnegie Mellon University
`aristesk@andrew.cmu.edu`

## Abstract

The vast majority of work in self-supervised learning have focused on assessing recovered features by a chosen set of downstream tasks. While there are several commonly used benchmark datasets, this lens of feature learning requires assumptions on the downstream tasks which are not inherent to the data distribution itself. In this paper, we present an alternative lens, one of parameter identifiability: assuming data comes from a parametric probabilistic model, we train a self-supervised learning predictor with a suitable parametric form, and ask whether the parameters of the optimal predictor can be used to extract the parameters of the ground truth generative model.

Specifically, we focus on latent-variable models capturing sequential structures, namely Hidden Markov Models with both discrete and conditionally Gaussian observations. We focus on masked prediction as the self-supervised learning task and study the optimal masked predictor. We show that parameter identifiability is governed by the task difficulty, which is determined by the choice of data model and the amount of tokens to predict. Technique-wise, we uncover close connections with the uniqueness of *tensor rank decompositions*, a widely used tool in studying identifiability through the lens of the method of moments.

## 1 Introduction

Self-supervised learning (SSL) is a relatively new approach to unsupervised learning, where the learning algorithm learns to predict auxiliary labels generated automatically from the data without human annotators. The hope is that with a properly designed prediction task, a successfully learned predictor would capture some knowledge about the underlying data. While SSL has been enjoying a rapid growth on the empirical front, theoretical understanding of why and when SSL works is still nascent. In no small part, this is because formalizing the desired guarantees seems challenging. For instance, the focus of SSL has largely been on learning *good features*, which in practice has been quantified by downstream performance on various benchmark datasets [Wang et al., 2018, 2019, Deng et al., 2009, Zhai et al., 2019, Tamkin et al., 2021]. To provide theoretical underpinning to this, one needs to make extra assumptions on the relationship between the self-supervised prediction task and the downstream tasks [Arora et al., 2019, Saunshi et al., 2020, HaoChen et al., 2021, Lee et al., 2021a, Wang et al., 2021, Wei et al., 2021, Wen and Li, 2021].

While associating SSL with downstream supervised tasks is a useful perspective and has led to several very interesting theoretical results, we take a step back and revisit a more general goal of SSL, which is to learn some informative functionals of the data distribution. Naturally, the key question here is

what functionals should be considered *informative*. While downstream performance is a notable valid choice, in this work, we choose an alternative criterion that is meaningful even without referencing any downstream tasks.

The alternative lens we are interested in is whether the functionals of the data distribution extracted by the SSL predictors can be simply stitched together to obtain the data distribution itself, given additional side-information about the family from which the data distribution is drawn. While this might seem like a tall order, masked prediction based SSL algorithms (which is essentially what pseudo-likelihood corresponds to) have classically been used for learning parametric graphical models such as Ising models [Ravikumar et al., 2010, Bresler, 2015, Vuffray et al., 2016]. But can this be done for broader classes of parametric models?

In this paper, we take a preliminary step towards this and ask the question of *parameter identifiability*: assuming the data comes from a ground truth parametric probabilistic model, can common self-supervised tasks uniquely identify the parameters of the ground truth model? More precisely, are the parameters of the model uniquely determined by the optimal predictor for the SSL task (Definition 1)? An appeal of this identifiability perspective is that when a SSL task is sufficient for parameter identifiability, the model parameters can then be recovered straightforwardly from the parameters from the optimal SSL predictor. Parameter identification also has the desirable property of being independent of any downstream task.

A priori, it is unclear whether we can achieve such model parameter identifiability via self-supervised tasks, since it requires recovering the full (parametric) generative model which is arguably more difficult than learning generic latent representations. This work provides a positive answer for broad classes of HMMs: we show that the commonly-used *masked prediction task* [Pathak et al., 2016, Devlin et al., 2018, He et al., 2021, Lee et al., 2021a], wherein a model is trained to predict a masked-out part of a sample given the rest of the sample, can identify the parameters of a HMM. As noted earlier, while such masked prediction for parameter learning has been applied in classical settings such as Ising models [Ravikumar et al., 2010, Bresler, 2015, Vuffray et al., 2016], the HMM setup in this work is more challenging due to the presence of latent variables. HMMs are also more suitable for modeling practical sequential data, and have been commonly adopted in theoretical analyses as a clean proxy for languages [Wei et al., 2021, Xie et al., 2021].

Concretely, the two HMM models we consider in this work are 1) the classic HMM with discrete latent and discrete observables, and 2) a HMM variant with discrete latents and continuous observables that are conditionally Gaussian given the latent, which we abbreviate as G-HMMs. We show that:

- Parameter identifiability is governed by the difficulty of the masked prediction task. The task difficulty is related to the amount of information provided by the combination of the model and the prediction task—where the difficulty can be increased by using a more complicated model, or by predicting more tokens. For instance, predicting the conditional mean of one token given another does not yield identifiability for a discrete HMM (Theorem 2), but does so when data comes from a G-HMM (Theorem 3). Moreover, the identifiability in the latter case quite strongly leverages structural properties of the posterior of the latent variables (Section 3.1).

- Tools for characterizing the uniqueness of tensor decompositions (e.g., Kruskal's Theorem [Kruskal, 1977, Allman et al., 2009]) can be leveraged to prove identifiability: For both HMM (Theorem 5) and G-HMM (Theorem 6), if we have predictors of the tensor product of tokens (e.g., $\mathbb{E}[x_2 \otimes x_3 | x_1]$), we can use the predictor output to construct a 3-tensor whose rank-1 components are uniquely determined and reveal the parameters of the model.

The rest of the paper is structured as follows. Section 2 provides relevant definitions, preliminaries and assumptions. Section 3 states the main results of this work. Main proofs, including the identifiability proof via tensor decomposition, are provided in Section 4, with the rest deferred to the appendix. We then discuss related works in Section 5. Finally, we emphasize that this work is a first-cut study on this lens of parameter recovery for analyzing SSL tasks based on masked prediction, and our encouraging results suggest interesting open directions in this thread of analyzing self-supervised learning via parameter recovery, which are discussed briefly in the conclusion.

## 2 Setup

This work focuses on two classes of latent-variable sequence models. The first are fully discrete hidden Markov models (HMMs), and the second are HMMs whose observables marginally follow a mixtures of Gaussians with identity covariance. We denote the observations and hidden states respectively by $\{x_t\}_{t\geq 1}$ and $\{h_t\}_{t\geq 1}$ for both classes. The hidden states $h_1 \to h_2 \to \cdots$ form a Markov chain, and conditional on $h_t$, the observable $x_t$ is independent of all other variables. Throughout, we refer to $\{x_t\}_{t\geq 1}$ as tokens, following the nomenclature from language models.

### 2.1 Models

**Discrete Hidden Markov Model**   We first describe the parameterization of the standard HMMs with discrete latents and observations. Let $\mathcal{X} := \{1, \ldots, d\} = [d]$ denote the observation space, and let $\mathcal{H} := [k]$ be the state space.[1] The parameters of interest are the *transition matrix* $T \in \mathbb{R}^{k \times k}$ and the *emission matrix* $O \in \mathbb{R}^{d \times k}$, defined in the standard way as

$$P(h_{t+1} = i \mid h_t = j) = T_{ij}, \qquad P(x_t = i \mid h_t = j) = O_{ij}.$$

**Conditionally-Gaussian HMM (G-HMM)**   We next describe the parameterization of *conditionally-Gaussian HMMs (G-HMMs)*. The state space $\mathcal{H} := [k]$ is the same as in the previous case, while the observation space is now continuous with $\mathcal{X} := \mathbb{R}^d$. The parameters of interest are $T \in \mathbb{R}^{k \times k}$, the transition matrix, and $\{\mu_i\}_{i \in [k]} \subset \mathbb{R}^d$, the means of the $k$ identity-covariance Gaussians. Precisely,

$$P(h_{t+1} = i \mid h_t = j) = T_{ij}, \qquad P(x_t = x \mid h_t = i) = (2\pi)^{-\frac{d}{2}} \exp\left(-\|x - \mu_i\|^2/2\right).$$

We use $M := [\mu_1, \ldots, \mu_k] \in \mathbb{R}^{d \times k}$ to denote the matrix whose columns are the Gaussian means.

### 2.2 Masked prediction tasks

We are interested in the (regression) task of predicting one or more "masked out" tokens as a function of another observed token, with the goal of minimizing expected squared loss under a distribution given by an HMM or G-HMM (equation 1). In the case of the discrete HMMs, we will specifically be predicting the *one-hot encoding vectors* of the observations. Thus, both for HMM and G-HMM, predicting a single token will correspond to predicting a vector. For notational convenience, we will simply associate the discrete states or observations via their one-hot vectors $\{e_1, e_2, \ldots\}$ in the appropriate space and interchangeably write $h = i$ or $h = e_i$, and similarly for $x$. For the task of predicting the tensor product of (one-hot encoding vectors of) tokens $\otimes_{\tau \in \mathcal{T}} x_\tau$ from another token $x_t$ (where $\mathcal{T}$ is some index set and $t \notin \mathcal{T}$), the optimal predictor with respect to the squared loss calculates the conditional expectation:

$$f(x_t) = \arg\min_{\tilde{f}} \mathbb{E}_{\{x_\tau\}_{\tau \in \mathcal{T}}} \|\text{vec}(\otimes_{\tau \in \mathcal{T}} x_\tau) - \text{vec}(\tilde{f}(x_t))\|_2^2 = \mathbb{E}[\otimes_{\tau \in \mathcal{T}} x_\tau \mid x_t] \in (\mathbb{R}^d)^{\otimes |\mathcal{T}|}, \quad (1)$$

where "vec" returns the vectorized form of a tensor.

We use the shorthand "$\otimes_{\tau \in \mathcal{T}} x_\tau | x_t$" to refer to this prediction task. For instance, consider the case of predicting $x_2$ given $x_1$ under the HMM with parameters $(O, T)$. The optimal predictor, denoted by $f^{2|1}$, can be written in terms of $(O, T)$ as [2]

$$f^{2|1}(x) = \mathbb{E}[x_2 \mid x_1 = x] = \sum_{i \in [k]} \mathbb{E}[x_2 \mid h_2 = i]P(h_2 = i \mid x_1 = x)$$

$$= \sum_{i \in [k]} \sum_{j \in [k]} \mathbb{E}[x_2 \mid h_2 = i]P(h_2 = i \mid h_1 = j)\underbrace{P(h_1 = j \mid x)}_{:=[\phi(x)]_j} = \sum_{i \in [k]} \sum_{j \in [k]} O_i T_{ij} \underbrace{\frac{O_{x,j}}{\sum_{l \in [k]} O_{x,l}}}_{:=[\phi(x)]_j}.$$

---

[1]Our results will assume $d \geq k$; see Section 2.3.

[2]The computation here relies on Assumption 1, given in Section 2.3.

Here $\phi : \mathbb{R}^d \to \mathbb{R}^k$ denotes the posterior distribution of a hidden state $h_t$ given the corresponding observation $x_t$, i.e., $\phi(x_t) = \mathbb{E}[h_t \mid x_t]$. [3]

Our goal is to study the parameter identifiability from the prediction tasks, when the predictors have the correct parametric form. Formally, we define identifiability from a prediction task as follows:

**Definition 1** (Identifiability from a prediction task, HMM). *A prediction task suffices for identifiability if, for any two HMMs with parameters $(O, T)$ and $(\tilde{O}, \tilde{T})$, equality of their optimal predictors for this task implies that there is a permutation matrix $\Pi$ such that $O = \tilde{O}\Pi$ and $T = \Pi^\top \tilde{T} \Pi$.*

In other words, the mapping from (the natural equivalence classes of) HMM distributions to optimal predictors for a task is injective, up to a permutation of the hidden state labels. By identifiability from a collection of prediction tasks, we refer to the injectiveness of the mapping from HMM distributions to the collections of optimal predictors for the tasks. Identifiability for G-HMMs is defined analogously with $O, \tilde{O}$ changed to $M, \tilde{M}$.

## 2.3  Assumptions

We now state the assumptions used in our results. The first assumption is that the transition matrices of the HMMs are doubly stochastic.

**Assumption 1** (Doubly stochastic transitions). *The transition matrix $T$ is doubly stochastic, and the marginal distribution of the initial hidden state $h_1$ is stationary with respect to $T$.*

This assumption guarantees that the stationary distribution of the latent distribution is uniform for any $t$, and the transition matrix for the reversed chain is simply $T^\top$. Moreover, this assumption reduces the parameter space and hence will make the non-identifiability results stronger.

We require the following conditions on the parameters for the discrete HMM:

**Assumption 2** (Non-redundancy, discrete HMM). *Every row of $O$ is non-zero.*

Assumption 2 can be interpreted as requiring each token to have a non-zero probability of being observed, which is a mild assumption. We also require the following non-degeneracy condition:

**Assumption 3** (Non-degeneracy, discrete HMM). $rank(T) = rank(O) = k \leq d$.

Note that Assumption 3 only requires the parameters to be non-degenerate, rather than have singular values bounded away from 0. The reason is that this work will focus on population level quantities and make no claims on finite sample behaviors or robustness.

For G-HMM, we similarly require the parameters to be non-degenerate:

**Assumption 4** (Non-degeneracy, G-HMM). $rank(T) = rank(M) = k \leq d$.

Moreover, we assume that the norms of the means are known and equal:

**Assumption 5** (Equal norms of the means). *For each $i \in [k]$, $\mu_i$ is a unit vector.*[4]

Assumptions 1-4 are fairly standard [see, e.g., Anandkumar et al., 2012]; in particular, Assumption 3, 4 are required to enable efficient learning, since learning degenerate HMMs can be computationally hard [Mossel and Roch, 2005]. Assumption 5 may be an artifact of our proofs, and it would be interesting to relax in future work.

Our notion of identifiability from a prediction task (or a collection of prediction tasks) will restrict attention to HMMs satisfying Assumptions 1, 2, 3 and G-HMMs satisfying Assumptions 1, 4, 5.

## 2.4  Uniqueness of tensor rank decompositions

Some of our identifiability results rely on the uniqueness of *tensor rank-1 decompositions* [Hitchcock, 1927]. An *order-t tensor* (or *t-tensor*) is an $t$-way multidimensional array; a matrix is a 2-tensor. The *tensor rank* of a tensor $W$ is the minimum number $R$ such that $W$ can be written as a sum of $R$

---

[3]For discrete HMMs, $\phi(x_t) = \frac{O^\top x_t}{\|O^\top x_t\|_1}$. For GHMMs, $[\phi(x_t)]_i = \frac{\exp\left(-\frac{\|x_t - \mu_i\|_2^2}{2}\right)}{\sum_{j \in [k]} \exp\left(-\frac{\|x_t - \mu_j\|_2^2}{2}\right)}, \forall i \in [k]$. $\phi$ does not need to be indexed by $t$ due to the stationarity assumption in Section 2.3.

[4]Assumption 5 can be changed to $\|\mu_i\|_2 = c$ for all $i \in [k]$, for any other fixed number $c > 0$.

rank-1 tensors. That is, if a $t$-tensor $W$ has rank-$R$, it means that $W = \sum_{i \in [R]} \otimes_{j \in [t]} U_i^{(j)}$ for some matrices $U^{(j)} \in \mathbb{R}^{n_j \times R}$, where $U_i^{(j)}$ denotes the $i^{\text{th}}$ column of matrix $U^{(j)}$.

In this work, we only need to work with 3-tensors of the form $W = \sum_{i \in [R]} A_i \otimes B_i \otimes C_i$ for some matrices $A \in \mathbb{R}^{n_1 \times R}$, $B \in \mathbb{R}^{n_2 \times R}$, $C \in \mathbb{R}^{n_3 \times R}$, as 3-tensors will suffice for identifiability in all of our settings of interest.[5] A classic work by Kruskal [1977] gives a sufficient condition under which $A, B, C$ can be recovered up to column-wise permutation and scaling. The condition is stated in terms of the *Kruskal rank*, which is the maximum number $r$ such that every $r$ columns of the matrix are linearly independent. Let $k_A$ denote the Kruskal rank of matrix $A$, then:

**Proposition 1** (Kruskal's theorem, Kruskal [1977])**.** *The components $A, B, C$ of a 3-tensor $W := \sum_{i \in [R]} A_i \otimes B_i \otimes C_i$ are identifiable up to a shared column-wise permutation and column-wise scaling if $k_A + k_B + k_C \geq 2R + 2$.*

We note that this work focuses on identifiability results rather than providing an algorithm or sample complexity bounds, though the proofs can be adapted into algorithms [see, e.g., Harshman, 1970] under slightly more restrictive conditions (which will be satisfied by all of our identifiability results).

## 3 Identifiability from masked prediction tasks

We now present the main (non-)identifiability results, and show that the combination of the data generative models and the prediction task directly impacts the sufficiency of identifiability.

### 3.1 Pairwise prediction

We begin with the simplest prediction task: namely predicting one token from another. We refer to such tasks as *pairwise prediction tasks*. For HMMs, this task fails to provide parameter identifiability:

**Theorem 2** (Nonidentifiability of HMM from predicting $x_t|x_1$)**.** *For any $t \in \mathbb{Z}, t \geq 2$, there exists a pair of HMM distributions with parameters $(O, T)$ and $(\tilde{O}, \tilde{T})$, each satisfying Assumptions 1, 2 and 3, such that the optimal predictors for the task $x_t|x_1$ are the same under each distribution, but there is no permutation matrix $\Pi \in \mathbb{R}^{k \times k}$ such that $\tilde{O} = O\Pi$ and $\tilde{T} = \Pi^\top T \Pi$ are both satisfied.*

Theorem 2 follows from the fact that the optimal predictor has the form of a product of (stochastic) matrices, and generally, one cannot uniquely recover matrices from their product sans additional conditions [Donoho and Elad, 2003, Candes et al., 2006, Spielman et al., 2012, Arora et al., 2014, Georgiev et al., 2005, Aharon et al., 2006, Cohen and Gillis, 2019]. Specifically, by equation 1, the optimal predictor is $f(x_1) = \mathbb{E}[x_t|x_1] = OT^{t-1}\phi(x_1)$ (where $\phi(x_1) := \mathbb{E}[h_1|x_t]$ is the posterior). When $t = 2$, we can find a non-permutation matrix $R$ such that $\tilde{O} = OR$, $\tilde{T} = R^\top TR$ give the same predictor as $O, T$. For $t > 2$, even if $\tilde{O} = O$, we show that the matrix power $T^{t-1}$ is not identifiable:

**Claim 1** (Nonidentifiability of matrix powers)**.** *For any $t \in \mathbb{Z}, t \geq 2$, there exist stochastic matrices $T, \tilde{T}$ satisfying Assumption 1, 3, such that $T \neq \tilde{T}$ and $T^t = \tilde{T}^t$.*

On the other hand, pairwise prediction actually *does* suffice for identifiability for G-HMM:

**Theorem 3** (Identifiability of G-HMM from predicting $x_2|x_1$)**.** *Under Assumption 1, 4, and 5, if the optimal predictors for the task $x_2|x_1$ under the G-HMM distributions with parameters $(M, T)$ and $(\tilde{M}, \tilde{T})$ are the same, then $(M, T) = (\tilde{M}, \tilde{T})$ up to a permutation of the hidden state labels.*

Comparing Theorem 2 and 3 shows that the specific parametric form of the generative model matters. Note that HMM and G-HMM have a similar form when conditioning on the latent variable; that is, with $t = 2$, the predictor conditioned on the hidden variable $h_2$ is $P(x_2|h_2 = i) = OT_i$ for HMM, and $P(x_2|h_2 = i) = MT_i$ for G-HMM. The salient difference between these two setups lies in the posterior function: while the posterior function for HMM is linear in the observable, the posterior function for G-HMM is more complicated and "reveals" more information about the parameter.

---

[5]To apply our results on higher order tensors, one can consider an order-3 slice of the higher order tensor.

To formalize the above intuition, first recall that the GHMM posterior has entries $[\phi(x_t)]_i = \frac{\exp\left(-\frac{\|x_t - \mu_i\|_2^2}{2}\right)}{\sum_{j \in [k]} \exp\left(-\frac{\|x_t - \mu_j\|_2^2}{2}\right)}$, $\forall i \in [k]$. We will show that for G-HMM, even matching the posterior function *nearly* suffices to identify $M$: if $M, \tilde{M}$ parameterize two posterior functions $\phi, \tilde{\phi}$ where $\phi = \tilde{\phi}$, then up to a permutation, $\tilde{M}$ must be equal to either $M$ or a unique (and somewhat special) transformation of $M$. The next step is to further exclude the (special) transformation, which is achieved using the constraint that $T, \tilde{T}$ are stochastic matrices. The first step of the proof sketch is captured by following lemma:

**Lemma 1.** *For $d \geq k \geq 2$, under Assumption 4, 5, $\phi = \tilde{\phi}$ implies $\tilde{M} = M$ or $\tilde{M} = HM$, where $H$ is a Householder transformation of the form $H := I_d - 2\hat{v}\hat{v}^\top \in \mathbb{R}^{d \times d}$, with $\hat{v} := \frac{(M^\dagger)^\top \mathbf{1}}{\sqrt{\mathbf{1}^\top M^\dagger (M^\dagger)^\top \mathbf{1}}}$.*

To provide some geometric intuition about how $H$ acts on $M$, note that $\hat{v}$ is a unit vector in the column space of $M$ and perpendicular to the affine hull of $\mathcal{A} := \{\mu_i : i \in [k]\}$, which means $\hat{v}^\top \mu_i$ is the same for all $i \in [k]$. As a result, $\tilde{M} = [\tilde{\mu}_1, ..., \tilde{\mu}_k] = [H\mu_1, ..., H\mu_k] = M - 2(\hat{v}^\top \mu_1)[\hat{v}, ..., \hat{v}]$ is a translation of $M$ along the direction of $\hat{v}$, such that the translated points $\{\tilde{\mu}_i\}_{i \in [k]}$ lie on the opposite side of the origin. It is non-trivial to argue that $HM$ is the only solution (other than $M$ itself) that preserves $\phi$, and we defer the proof to Appendix A.1.2. It is, however, easy to see that $HM$ indeed results in a matching posterior, whose sufficient conditions are 1) $\tilde{M}$ is a translation of $M$, and 2) $\|\tilde{\mu}_i\|^2 - \|\mu_i\|^2$ is the same for all $i \in [k]$. $\tilde{M} := HM$ indeed satisfies both conditions.

*Proof sketch for Theorem 3*: We first show that if $M, T$ and $\tilde{M}, \tilde{T}$ produce the same predictor, then their posterior function must be equal up to a permutation (Lemma 2). We can then apply Lemma 1 to recover $M$ up to a permutation and a Householder transformation $H$. Then, we show that if $\tilde{M} = HM$, then the corresponding $\tilde{T}$ must have negative entries and thus would not be a valid stochastic matrix. Hence it must be that $\tilde{M}, M$ are equal up to permutation.

Finally, by way of remarks, another way to think of the difference between the two setups is that for HMM, $P(x_2|x_1)$ is a mixture of categorical distributions, which itself is also a categorical distribution. This also implies that the nonidentifiability from pairwise prediction in the HMM case cannot be resolved by changing the squared loss to another proper loss function. On the other hand, for G-HMM, the conditional distribution $P(x_2|x_1)$ is a mixture of Gaussians, which is well known to be identifiable. In fact, if we were given access to the *entire* conditional distribution $P(x_2|x_1)$ (instead of just the conditional mean), it is even easier to prove identifiability for G-HMM. Though this is already implied from identifiability from the conditional means, we provided a (much simpler) proof in Appendix A.3 assuming access to the full conditional distribution.

## 3.2 Beyond pairwise prediction

The conclusion from Theorem 2 is that a single pairwise prediction task does not suffice for identifiability on HMMs. The next question is then: can we modify the task to obtain identifiability? A natural idea is to force the model to "predict more", and one straightforward way to do so is to combine multiple pairwise prediction tasks. It turns out that this does not resolve the nonidentifiability issue, as we can show that the parameters are not identifiable even when considering *all* possible pairwise tasks involved 3 (adjacent) tokens:

**Theorem 4** (Nonidentifiability of HMM from all pairwise predictions on 3 tokens)**.** *There exists a pair of HMM distributions with parameters $(O, T)$ and $(\tilde{O}, \tilde{T})$, each satisfying Assumptions 1, 2 and 3, and also $\tilde{O} \neq O$, such that, for each of the tasks $x_2|x_1$, $x_1|x_2$, $x_3|x_1$, and $x_1|x_3$, the optimal predictors are the same under each distribution.*[6]

We briefly remark that the reason for only considering adjacent time steps is that when the tokens are at least two time steps apart, matching predictors only matches powers of the transition matrices, which in general does not ensure the transition matrices themselves are matched as shown in Claim 1.

---

[6]These 4 pairwise tasks cover all possible pairwise tasks on 3 adjacent tokens. In particular, there is no need to consider $x_2|x_3$ or $x_3|x_2$, since they are the same as $x_1|x_2$ and $x_2|x_1$.

For the intuition of the nonidentifiability result in Theorem 4, recall that the limitation of pairwise predictions on HMMs comes from non-uniqueness of matrix factorization. While adding additional pairwise prediction tasks introduces more equations on the product of matrices, these equations are highly dependent, and the proof works by providing counterexamples that can simultaneously satisfy all these equations.

The above intuition leads to another way of forcing the model to "predict more", that is, to increase the number of predicted tokens. The hope is that doing so results in equations on tensors—as opposed to matrices— for which there is a lot of classical machinery delineating tensors for which the rank-1 decomposition is unique, as discussed in Section 2.4. This intuition proves to be true and we show that increasing the number from 1 to 2 already suffices for identifiability:

**Theorem 5** (Identifiability from masked prediction on three tokens, HMM). *Let $(t_1, t_2, t_3)$ be any permutation of $(1, 2, 3)$, and consider the prediction task $x_{t_2} \otimes x_{t_3} | x_{t_1}$. Under Assumption 1, 2, 3, if the optimal predictors under the HMM distributions with parameters $(O, T)$ and $(\tilde{O}, \tilde{T})$ are the same, then $(O, T) = (\tilde{O}, \tilde{T})$ up to a permutation of the hidden state labels.*

Compared to prior results on identifiability from third order moments [Allman et al., 2009, Anand-kumar et al., 2012, 2014], the difficulty in our setup is that we only have access to the conditional 2-tensors (i.e. matrices) given by the predictors. The proof idea is to construct a third-order tensor by linearly combining the conditional 2-tensors for each possible value of the token being conditioned on, such that Kruskal's theorem applies and gives identifiability. Note, importantly, that the weights for the linear combination cannot depend on the marginal probabilities of the token being conditioned on, since we do not have access to these marginals, and it is unclear whether we could extract unique marginals given the conditional probabilities we are predicting. Thus, the above theorem cannot be simply derived from results showing parameter identifiability from the 3rd order moments.

It can be show that this tensor decomposition argument can also be applied to G-HMM, with the help of Lemma 1. We leave the details to Theorem 6 in Appendix A.

# 4 Proofs

We now discuss proofs for some of the main results. Section 4.1 proves the identifiability of HMM parameters from the task of predicting two tokens (Theorem 5) using ideas from tensor decomposition, and Section 4.2 shows the identifiability proof of pairwise prediction on G-HMM. The rest of the proofs are deferred to the appendix.

## 4.1 Proof of Theorem 5: identifiability of predicting two tokens for HMM

There are three cases for the two-token prediction task, i.e. 1) $x_2 \otimes x_3 | x_1$, 2) $x_1 \otimes x_3 | x_2$, and 3) $x_1 \otimes x_2 | x_3$. We will prove for the first two cases, as the third case is proved the same way as the first case by symmetry. In all cases, the idea is to use the predictor to construct a 3-tensor whose components are each of rank-$k$, so that applying Kruskal's theorem gives identifiability.

**Case 1, $x_2 \otimes x_3 | x_1$:** $O, T$ and $\tilde{O}, \tilde{T}$ producing the same predictor means $f^{2\otimes3|1}(x_1) := \mathbb{E}[x_2 \otimes x_3 | x_1] = \tilde{\mathbb{E}}[x_2 \otimes x_3 | x_1] := \tilde{f}^{2\otimes3|1}(x_1)$, where $\mathbb{E}, \tilde{\mathbb{E}}$ are parameterized by the corresponding parameters. Let $\mathcal{X} := \{e_i : i \in [d]\}$, and consider the following 3-tensor:

$$
\begin{aligned}
W &:= \sum_{x_1 \in \mathcal{X}} x_1 \otimes \mathbb{E}[x_2 \otimes x_3 | x_1] = \sum_{x_1 \in \mathcal{X}} x_1 \otimes \mathbb{E}_{h_2|x_1}[\mathbb{E}[x_2 \otimes x_3 | x_1]|h_2] \\
&= \sum_{i \in [k]} \sum_{x_1 \in \mathcal{X}} P(h_2 = i | x_1) x_1 \otimes \mathbb{E}[x_2 | h_2 = i] \otimes \mathbb{E}[x_3 | h_2 = i] \\
&= \sum_{i \in [k]} \Big( \underbrace{\sum_{x_1 \in \mathcal{X}} (T\phi(x_1))^\top e_i^{(k)} x_1}_{:= a_i} \Big) \otimes O_i \otimes (OT)_i,
\end{aligned}
\tag{2}
$$

where $O_i$ denotes the $i^{\text{th}}$ column of $O$, and similarly for $(OT)_i$. Note that $W$ can also be written as

$$W = \sum_{x_1 \in \mathcal{X}} x_1 \otimes \tilde{\mathbb{E}}[x_2 \otimes x_3 | x_1] = \sum_{i \in [k]} \Big( \sum_{x_1 \in \mathcal{X}} (\tilde{T}\tilde{\phi}(x_1))^\top e_i^{(k)} x_1 \Big) \otimes \tilde{O}_i \otimes (\tilde{O}\tilde{T})_i. \tag{3}$$

We want to apply Kruskal's theorem for identifiability. In particular, we will show that each component in equation 2 forms a matrix of Kruskal rank $k$. The second and third components clearly satisfy this condition by Assumption 3. For the first component, recall that $\phi(x) = \frac{O^\top x}{\|O^\top x\|_1}$ and write $a_i$ as

$$a_i = \sum_{j \in [d]} \big(T\phi(e_j^{(d)})\big)^\top e_i^{(k)} \cdot e_j^{(d)} = \text{diag}\left([\frac{1}{\|(e_j^{(d)})^\top O\|_1}]_{j \in [d]}\right) OT^\top e_i^{(k)}. \tag{4}$$

Putting $a_i$ into a matrix form, we get $A := [a_1, ..., a_k] = \text{diag}\big([1/\|(e_j^{(d)})^\top O\|_1]_{j \in [d]}\big) OT^\top,$ [7] which is of rank $k$ by Assumption 3. Hence components $W$ are all of Kruskal rank $k$, and columns of $OT, O$ are identified up to column-wise permutation and scaling by Kruskal's theorem. The indeterminacy in scaling is further removed noting that columns of $O, T$ need to sum up to 1. Lastly, $T$ is recovered as $T = O^\dagger OT$.

**Case 2, $x_1 \otimes x_3 | x_2$:** The optimal predictor for the task of predicting $x_1, x_3$ given $x_2$ takes the form

$$\mathbb{E}[x_1 \otimes x_3 | x_2] = (OT^\top)\text{diag}(\phi(x_2))(OT)^\top. \tag{5}$$

Similarly as the previous case, we would like to construct a 3-tensor whose components can be uniquely determined by Kruskal's theorem. Let $\mathcal{X}$ be the same as before, and consider the 3-tensor

$$W := \sum_{x_2 \in \mathcal{X}} x_2 \otimes \mathbb{E}[x_1 \otimes x_3 | x_2] = \sum_{x_2 \in \mathcal{X}} x_2 \otimes \mathbb{E}_{h_2 | x_2}(\mathbb{E}[x_1 | h_2] \otimes \mathbb{E}[x_3 | h_2])$$

$$= \sum_{i \in [k]} \underbrace{\sum_{x_2 \in \mathcal{X}} (\phi(x_2))^\top e_i^{(k)} x_2}_{:=a_i} \otimes \mathbb{E}[x_1 | h_2] \otimes \mathbb{E}[x_3 | h_2] = \sum_{i \in [k]} a_i \otimes (OT^\top)_i \otimes (OT)_i, \tag{6}$$

where the first component can be simplified to

$$a_i = \sum_{j \in [d]} \frac{(e_j^{(d)})^\top O}{\|(e_j^{(d)})^\top O\|_1} e_i^{(k)} \cdot e_j^{(d)} = \Big(\text{diag}\big([\|O_j^\top\|_1]_{j \in [d]}\big)\Big)^{-1} O e_i^{(k)} := D^{-1} O e_i^{(k)}. \tag{7}$$

The matrix $A := [a_1, ..., a_k] = D^{-1}O$ is of rank $k$, hence we can identify (up to permutation) columns of each component of $W$ by Kruskal's theorem. This means if $O, T$ and $\tilde{O}, \tilde{T}$ produce the same predictor, then we have $OT = \tilde{O}\tilde{T}, OT^\top = \tilde{O}\tilde{T}^\top$, and that $O, \tilde{O}$ are matched up to a scaling of rows (i.e. $D^{-1}$). Next, to determine $D$, note that $T, \tilde{T}$ are doubly stochastic by Assumption 1, which means the all-one vector $\mathbf{1} \in \mathbb{R}^k$ satisfies $T\mathbf{1} = \tilde{T}\mathbf{1} = \mathbf{1}$. Hence $\tilde{O}\tilde{T}\mathbf{1} = OT\mathbf{1} = O\mathbf{1} = [\|O_j^\top\|_1]_{j \in [d]}$. We can then compute $D$ as $D = \text{diag}(OT\mathbf{1})$, and recover $O$ as $O = DA$. Finally, $T$ is also recovered since $\tilde{O}\tilde{T} = O\tilde{T} = OT \Rightarrow \tilde{T}T^{-1} = I_k \Rightarrow \tilde{T} = T$.

### 4.2 Proof of Theorem 3: identifiability of predicting $x_2$ given $x_1$ for G-HMM

For G-HMM, the predictor for $x_2$ given $x_1$ is parameterized as $f^{2|1}(x_1) = \mathbb{E}[x_2 | x_1] = MT\phi(x_1)$. If $M, T$ and $\tilde{M}, \tilde{T}$ produce the same predictor, then

$$f^{2|1}(x) = MT\phi(x) = \tilde{M}\tilde{T}\tilde{\phi}(x) = \tilde{f}^{2|1}(x), \forall x \in \mathbb{R}^d. \tag{8}$$

Let $R := (\tilde{M}\tilde{T})^\dagger(MT) \in \mathbb{R}^{k \times k}$, then $\tilde{\phi}(x) = R\phi(x)$. The following lemma (proof deferred to Appendix A.1) says that $\phi, \tilde{\phi}$ must then be equal up to a permutation of coordinates:

**Lemma 2.** *If there exists a non-singular matrix $R \in \mathbb{R}^{k \times k}$ such that $\phi(x) = R\tilde{\phi}(x), \forall x \in \mathbb{R}^d$, then $R$ must be a permutation matrix.*

---

[7] We use $[\alpha_i]_{i \in [d]}$ to denote a $d$-dimensional vector whose $i^{\text{th}}$ entry is $\alpha_i$.

Combined with Lemma 1, we have $\tilde{M}$ is equal to (up to a permutation) either $M$ or $HM$, where $H$ is the Householder reflection given in Lemma 1.

The remaining step is to show that $HM$ can be ruled out by requiring $\tilde{T}$ to be a stochastic matrix. Note that matching both the predictor and the posterior function means we also have $\tilde{M}\tilde{T} = MT$, or $\tilde{T} = (\tilde{M}^\dagger M)T$. Recall that $H := I_d - 2\hat{v}\hat{v}^\top$ for $\hat{v} = \frac{(M^\dagger)^\top \mathbf{1}}{\sqrt{\mathbf{1}^\top M^\dagger (M^\dagger)^\top \mathbf{1}}}$. When $\tilde{M} = H\tilde{M}$, the column sum of $\tilde{M}^\dagger M$ is

$$\mathbf{1}^\top \tilde{M}^\dagger M = \mathbf{1}^\top M^\dagger H^{-1} M = \mathbf{1}^\top M^\dagger (I - 2\hat{v}\hat{v}^\top) M = \mathbf{1}^\top (I - 2M^\dagger \hat{v}\hat{v}^\top M)$$
$$= \mathbf{1}^\top - 2 \cdot \mathbf{1}^\top \frac{M^\dagger (M^\dagger)^\top \mathbf{1}\mathbf{1}^\top M^\dagger M}{\mathbf{1}^\top M^\dagger (M^\dagger)^\top \mathbf{1}} = \mathbf{1}^\top - 2 \cdot \frac{\mathbf{1}^\top M^\dagger (M^\dagger)^\top \mathbf{1}}{\mathbf{1}^\top M^\dagger (M^\dagger)^\top \mathbf{1}} \mathbf{1}^\top = \mathbf{1}^\top - 2 \cdot \mathbf{1}^\top = -\mathbf{1}^\top. \quad (9)$$

This means the column sum of $\tilde{T}$ is $\mathbf{1}^\top \tilde{T} = \mathbf{1}^\top (\tilde{M}^\dagger M)T = -\mathbf{1}^\top T = -\mathbf{1}^\top$, which violates the constraint that $\tilde{T}$ should be a stochastic matrix with positive entries and column sum 1. Hence it must be that $M = \tilde{M}$ and hence also $T = \tilde{T}$ (up to permutation), proving the theorem statement.

## 5 Related works

**Self-supervised learning** On the empirical side, self-supervised methods have gained a great amount of popularity across many domains, including language understanding [Mikolov et al., 2013, Vaswani et al., 2017, Devlin et al., 2018], visual understanding [Doersch et al., 2015, Pathak et al., 2016], and distribution learning [Gutmann and Hyvärinen, 2010, Gao et al., 2020]. Classic ideas such as contrastive learning [Hadsell et al., 2006, Gutmann and Hyvärinen, 2010, Dosovitskiy et al., 2014] and masked prediction [Mikolov et al., 2013] remain powerful in their modern realizations [Hénaff et al., 2019, Chen et al., 2020b, Devlin et al., 2018, Radford et al., 2019, Chen et al., 2020a, He et al., 2021], pushing the state of the art performance and even surpassing supervised pretraining in various aspects [Lee et al., 2021b, Liu et al., 2021].

On the theoretical front, there have been analyses on both masked predictions [Lee et al., 2021a, Zhang and Hashimoto, 2021] and contrastive methods [Arora et al., 2019, Tosh et al., 2020a,b, Wang and Isola, 2020, HaoChen et al., 2021, Wen and Li, 2021], with a focus on characterizing the quality of the learned features for downstream tasks [Saunshi et al., 2020, Wei et al., 2021]. These approaches usually rely on quite strong assumptions to tie the self-supervised learning objective to the downstream tasks of interest. In contrast, our work takes the view of parameter identifiability, for which there is no need for downstream assumptions but instead the specific parametric form is key. Note also that while the parameter recovery lens is a new contribution of our work, Wen and Li [2021] argue (as a side-product of their analysis) that some aspects of a generative model are recovered in their setup. Their data model, however, is substantially different from ours and has very different identifiability properties (owing to its basis in sparse coding).

**Latent variable models and tensor methods** Latent variable models have been widely studied in the literature. One important area of research is independent component analysis (ICA), where the data is assumed to be given as a transformation (mixing) of unknown independent sources which ICA aims to identify. In nonlinear ICA data models, both the sources and the mixing function are generally not identifiable. However, identifiability of the sources can be shown under some additional assumptions (e.g. on the dependency structure of different time steps) [Hyvarinen and Morioka, 2016, 2017, Hälvä and Hyvarinen, 2020]. Similar ideas have also been applied in the self-supervised setting, where the latent variables can be identified under suitable assumptions on the conditional distribution of the latent [Zimmermann et al., 2021] or on data augmentations [Von Kügelgen et al., 2021]. Unlike our setup though, the mixing function in these models is deterministic and not the object of recovery.

More related to this work is the line of work on learning latent variable models with tensor methods. Specific to learning HMMs, Mossel and Roch [2005] and Anandkumar et al. [2012, 2014] provide algorithms based on third-order moments. A major difference between these prior works on tensor methods and ours is that previous results operate on joint moments, while the results in this work are based on conditional moments given by the optimal predictors for the masked tokens.

# 6  Conclusion

In this work, we take a model parameter identifiability view of self-supervised learning, which offers a complementary perspective to the current focus of feature learning for downstream performance. By analyzing the masked prediction task in the setup of HMMs and its conditionally-Gaussian variant G-HMM, we showed that parameter recovery is determined by the task difficulty, which can be tuned by both changing the parametric form of the data generative model, and by changing the masked prediction task.

We emphasize that this is a first-cut effort in the research program of analyzing SSL through the lens of model identifiability; we aim to build on this foundation to extend our analyses from HMMs to more complicated latent sequence and latent variable models. We also note that we have focused here on population analyses, and model identifiability. It would be of interest to build off this to develop and analyze the corresponding finite-sample learning algorithms for parametric generative models given SSL tasks, with sample complexity results, both in the realizable case, as well as in the agnostic case where we have model mis-specification. Given the use of conditional MLEs and regressions in SSL, and the natural robustness of these to model mis-specifications, we conjecture that these approaches will be much more robust when compared to say spectral methods.

Overall, we hope this work on an alternative lens to analyze SSL inspires further research.

## Acknowledgement

The authors would like to thank Elan Rosenfeld for insightful discussions and comments. We acknowledge the support of ONR via N000141812861, NSF via IIS-1909816, IIS-1955532, IIS-2211907, IIS-1563785, a JP Morgan Faculty Award, and the CMU/PwC DT&I Center.

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
