# A Missing proofs for G-HMM

This section provides missing proofs for results on G-HMM. We will first prove the two lemmas on properties of the posterior function (Lemma 1, 2), then show the proof for the three-token prediction task (Theorem 6) using a tensor decomposition idea similar to that of Theorem 5. At the end, we show the identifiability from pairwise conditional distributions (as opposed to conditional expectation as in masked prediction tasks), which is proved by reducing parameter recovery to the identifiability of Gaussian mixtures (Theorem 7).

## A.1 Proofs of helper lemmas

### A.1.1 Proof for Lemma 2

Given the form of the predictor, matching two predictors $f, \tilde{f}$ means that the corresponding posteriors $\phi, \tilde{\phi}$ are matched up to a linear transformation. We will now prove the following lemma, which says that in this case, $\phi, \tilde{\phi}$ can in fact only differ by a permutation of coordinates:

**Lemma** (Lemma 2 restated). *If there exists a non-singular matrix $R \in \mathbb{R}^{k \times k}$ such that $\phi(x) = R\tilde{\phi}(x)$, $\forall x \in \mathbb{R}^d$, then $R$ must be a permutation matrix.*

*Proof.* We will prove the lemma by matching the Jacobian w.r.t. $x$ on both sides. Let's first quickly recall the Jacobian of the posterior vector $\phi(x) \in \mathbb{R}^k$, where $[\phi(x)]_i = \frac{\exp(-\frac{\|x-\mu_i\|^2}{2})}{\sum_{j \in [k]} \exp(-\frac{\|x-\mu_j\|^2}{2})}$. Denote $o(x) := \left[ -\frac{\|x-\mu_1\|^2}{2}, ..., -\frac{\|x-\mu_k\|^2}{2} \right] \in \mathbb{R}^k$, then $\nabla_x \phi(x) = \nabla_{o(x)} \text{softmax}(o(x)) \cdot \nabla_x o(x)$, where

$$\nabla_o [\text{softmax}(o)]_i = [\text{softmax}(o)]_i \cdot (e_i - \text{softmax}(o)) = [\phi(x)]_i \cdot (e_i - \phi(x)),$$
$$\nabla_o \text{softmax}(o) = \text{diag}(\phi(x)) - \phi(x)\phi(x)^\top, \tag{10}$$
$$\nabla_x o(x) = -[x - \mu_1, ..., x - \mu_k]^\top.$$

Hence the Jacobian is

$$\nabla_x \phi(x) = \left( \text{diag}(\phi(x)) - \phi(x)\phi(x)^\top \right) \cdot (M - [x, x, ..., x])^\top. \tag{11}$$

Denote $\Delta := M - [x, x, ..., x] \in \mathbb{R}^{d \times k}$, and similarly $\tilde{\Delta} = \tilde{M} - [x, x, ..., x]$. Matching $\nabla_x \tilde{\phi}(x) = \nabla_x R\phi(x)$ gives

$$\text{diag}(R\phi(x))\tilde{\Delta}^\top - R\phi(x)(\tilde{\Delta}R\phi(x))^\top = R\text{diag}(\phi(x))\Delta^\top - R\phi(x)(\Delta\phi(x))^\top. \tag{12}$$

Let's take $x = x_c^{(i)} := c\mu_i$ for $c > 1$. We claim that this $x_c^{(i)}$ satisfies $\lim_{c \to \infty} \phi(x_c^{(i)}) \to e_i$. This is because $\forall j \neq i$,

$$\lim_{c \to \infty} \frac{[\phi(x_c^{(i)})]_j}{[\phi(x_c^{(i)})]_i} = \lim_{c \to \infty} \exp\left( \frac{\|c\mu_i - \mu_i\|^2}{2} - \frac{\|c\mu_i - \mu_j\|^2}{2} \right)$$
$$= \lim_{c \to \infty} \exp\left( -\frac{((2c-1)\mu_i - \mu_j)^\top (\mu_i - \mu_j)}{2} \right) = \lim_{c \to \infty} \exp\left( -\frac{2c\mu_i^\top(\mu_i - \mu_j)}{2} \right) = 0 \tag{13}$$

where the last equality is because $\mu_i^\top(\mu_i - \mu_j) > 0$ for any $\mu_i, \mu_j$ lying on the same hypersphere.

With such choices of $x$, the two sides of equation 12 are now:

$$LHS = \text{diag}(R_i)\tilde{\Delta}^\top - R_i R_i^\top \tilde{\Delta}^\top = (\text{diag}(R_i) - R_i R_i^\top)\tilde{\Delta}^\top$$
$$= RHS = \sum_{j \in [k]} [e_i]_j R_j(\Delta_j)^\top - Re_i(\Delta e_i)^\top = R_i(\Delta_i)^\top - R_i(\Delta_i)^\top = 0. \tag{14}$$

Since $x := c\mu_i$ for $c \to \infty$ lies outside the affine hull of $\{\tilde{\mu}_i\}_{i \in [k]}$, $\tilde{\Delta}$ is of full rank due to the following claim:

**Claim 2.** *Given a linearly independent set $\{u_i\}_{i\in[k]}$, if $\{u_i - v\}_{i\in[k]}$ is not linearly independent, then $v = \sum_{i\in[k]} \beta_i \cdot u_i$ where $\sum_{i\in[k]} \beta_i = 1$.*

*Proof.* Since $\{u_i - v\}_{i\in[k]}$ is linearly dependent, we can write some $u_j - v$ as the linear combination of other $\{u_i - v\}_{i\in[k], i\neq j}$. Let's take $j = k$ wlog, and denote the coefficients of the linear combination as $\{\alpha_i\}_{i\in[k-1]}$. Then

$$u_k - v = \sum_{i\in[k-1]} \alpha_i(u_i - v) \Rightarrow \big(1 - \sum_{i\in[k-1]} \alpha_i\big)v = - \sum_{i\in[k-1]} \alpha_i \cdot u_i + u_k \tag{15}$$

The right hand side is non-zero since $\{u_i\}_{i\in[k]}$ are linearly independent by assumption, hence $1 - \sum_{i\in[k-1]} \alpha_i \neq 0$, and we get

$$v = \sum_{i\in[k-1]} \underbrace{\frac{-\alpha_i}{1 - \sum_{i\in[k-1]}\alpha_i}}_{:=\beta_i} \cdot u_i + \underbrace{\frac{1}{1 - \sum_{i\in[k-1]}\alpha_i}}_{:=\beta_k} u_k. \tag{16}$$

Note that $\sum_{i\in[k]} \beta_i = 1$, hence $v$ is an affine combination of $\{u_i : i \in [k]\}$. $\square$

Since $\tilde{\Delta}$ is full rank, it must be $\mathrm{diag}(R_i) - R_i R_i^\top = 0$, which implies $R$ is a permutation matrix. This is because for any non-zero $v$ s.t. $\mathrm{diag}(v) - vv^\top = 0$, the entries of $v$ satisfy $v_i^2 = 1$, $v_i v_j = 0$ for $i \neq j$. Hence $v$ has exactly one non-zero entry which is $\pm 1$. Since $R\phi(x) = \tilde{\phi}(x)$ where $\phi(x), \tilde{\phi}(x)$ are both probability vectors with non-negative entries, this non-zero entry has to be 1 (and not -1). Since $R$ is of rank-$k$ by Assumption 4, this non-zero entry is at different positions for different $R_i$, hence $R$ is a permutation matrix.

$\square$

### A.1.2 Proof of Lemma 1

We show that if $M, \tilde{M}$ parameterize $\phi, \tilde{\phi}$ respectively and that $\phi = \tilde{\phi}$, then $\tilde{M}$ must equal to either $M$ or a unique (and somewhat special) transformation of $M$:

**Lemma** (Lemma 1 restated). *For $d \geq k \geq 2$, then under Assumption 4, 5, $\phi = \tilde{\phi}$ implies $\tilde{M} = M$ or $\tilde{M} = HM$, where $H$ is a Householder transformation of the form $H := I_d - 2\hat{v}\hat{v}^\top \in \mathbb{R}^{d\times d}$, with $\hat{v} := \frac{(M^\dagger)^\top \mathbf{1}}{\sqrt{\mathbf{1}^\top M^\dagger (M^\dagger)^\top \mathbf{1}}}$.*

*Proof.* Let's start with $d = k$. First, let's check the conditions for $\phi = \tilde{\phi}$. For any $x \in \mathbb{R}^d$, we have

$$[\phi(x)]_i = \frac{\exp\big(-\frac{\|x-\mu_i\|^2}{2}\big)}{\sum_{j\in[k]}\exp\big(-\frac{\|x-\mu_j\|^2}{2}\big)} = \frac{\exp\big(-\frac{\|x-\tilde{\mu}_i\|^2}{2}\big)}{\sum_{j\in[k]}\exp\big(-\frac{\|x-\tilde{\mu}_j\|^2}{2}\big)} = [\tilde{\phi}(x)]_i, \ \forall i \in [k]$$

$$\Rightarrow \frac{\exp\big(-\frac{\|x-\mu_i\|^2}{2}\big)}{\exp\big(-\frac{\|x-\tilde{\mu}_i\|^2}{2}\big)} = \frac{\exp\big(-\frac{\|x-\mu_j\|^2}{2}\big)}{\exp\big(-\frac{\|x-\tilde{\mu}_j\|^2}{2}\big)}, \forall i,j \in [k] \tag{17}$$

$$\Rightarrow \|x - \mu_i\|^2 - \|x - \tilde{\mu}_i\|^2 = \|x - \mu_j\|^2 - \|x - \tilde{\mu}_j\|^2, \ \forall i,j \in [k]$$

$$\Rightarrow 2\big((\tilde{\mu}_i - \mu_i) - (\tilde{\mu}_j - \mu_j)\big)^\top x = \big(\|\mu_j\|^2 - \|\tilde{\mu}_j\|^2\big) - \big(\|\mu_i\|^2 - \|\tilde{\mu}_i\|^2\big), \ \forall i,j \in [k].$$

Since the left hand side is linear in $x \in \mathbb{R}^d$ and the right hand side is a constant, it must be that both sides are 0. That is, the necessary conditions for $\phi = \tilde{\phi}$ are that for any $i, j \in [k]$, 1) $\tilde{\mu}_i - \mu_i = \tilde{\mu}_j - \mu_j$, and 2) $\|\mu_i\|^2 - \|\tilde{\mu}_i\|^2 = \|\mu_j\|^2 - \|\tilde{\mu}_j\|^2$. It can be checked that these two conditions are also sufficient for $\phi = \tilde{\phi}$.

Denote $v := \mu_i - \tilde{\mu}_i$. The norms of the means are known and equal by Assumption 5, which gives

$$\|\mu_i\|^2 - \|\tilde{\mu}_i\|^2 = \|\mu_i\|^2 - \|\mu_i - v\|^2 = (2\mu_i - v)^\top v = 0, \ \forall i \in [k]. \tag{18}$$

The last equality in equation 18 holds for a non-zero $v$ when the span of $\{2\mu_i - v : i \in [k]\}$ is $(d-1)$-dimensional subspace. On the other hand, the span of $\{2\mu_i - v : i \in [k]\}$ is at least $(k-1)$ by Assumption 4. When $d = k$, it must be that the dimension is exactly $(d-1)$, which means $v$ is an affine combination of $\{2\mu_i : i \in [k]\}$ by Claim 2.

Moreover, $v$ has to be orthogonal to $\{2\mu_i - v : i \in [k]\}$, which leads to the unique choice of $v$ that is the projection of the origin onto the $(d-1)$-dimensional subspace specified by the affine combinations of $\{2\mu_i : i \in [k]\}$.

**Claim 3.** *$v$ is the projection of the origin to the hyperplane defined by $\{2\mu_i : i \in [k]\}$, and is the only solution to equation 18.*

*Proof.* It is clear that this choice of $v$ satisfies $(2\mu_i - v)^\top v = 0, \forall i \in [k]$. To see that this is the unique choice, suppose there exists some $v'$ lying in the hyperplane of $\{2\mu_i\}$, and denote $\delta := v' - v$.

Note that $\delta^\top v = 0$: let the hyperplane specified by $\{2\mu_i\}_{i\in[k]}$ be specified as $\{x : \langle u, x \rangle = c\}$ for some $u \in \mathbb{R}^d$ and $c \in \mathbb{R}$. Then $v$, the projection of the origin, can be written as $v = \frac{c}{\|u\|} \cdot \frac{u}{\|u\|}$, i.e. $v$ is proportional to the normal vector $u$. For any $v'$ in the hyperplane, it satisfy $\langle u, v' \rangle = c$, and

$$
\begin{aligned}
\delta^\top v &= (v' - v)^\top v = \langle \frac{c}{\|u\|} \frac{u}{\|u\|}, v' \rangle - \left\| \frac{c}{\|u\|} \frac{u}{\|u\|} \right\|^2 \\
&= \frac{c}{\|u\|^2} \cdot \langle u, v' \rangle - \frac{c^2}{\|u\|^2} \frac{\|u\|^2}{\|u\|^2} = \frac{c^2}{\|u\|^2} - \frac{c^2}{\|u\|^2} = 0.
\end{aligned}
\tag{19}
$$

Then for any $v'$ satisfying equation 18,

$$
\begin{aligned}
(2\mu_i - v')^\top v' &= (2\mu_i - v - \delta)^\top (v + \delta) \\
&= \underbrace{(2\mu_i - v)^\top v}_{0} + 2\mu_i^\top \delta - \underbrace{v^\top \delta}_{0} - \underbrace{\delta^\top v}_{0} - \delta^\top \delta = (2\mu_i - \delta)^\top \delta = 0, \ \forall i \in [k].
\end{aligned}
\tag{20}
$$

Since $\{2\mu_i - \delta\}_{i\in[k]}$ spans the $(k-1)$-dimensional hyperplane and that $\delta$ lies in the hyperplane, it must be that $\delta = 0$, i.e. $v' = v$. $\qquad\square$

Note that this choice of $v$ also satisfies $\|\mu_i - v\| = \|\mu_i\|$, since $v$ and the origin are reflections w.r.t. the hyperplane that is the affine hull of $\{\mu_i : i \in [k]\}$. In other words, $\{\mu_i - v\}_{i\in[k]}$ is related to $\{\mu_i\}_{i\in[k]}$ via the Householder transformation of the form $H := I_d - 2\frac{vv^\top}{\|v\|^2}$, i.e. $\mu_i - v = H\mu_i$. Denote $\hat{v} := \frac{v}{\|v\|_2}$. An explicit formula for $\hat{v}$ is $\hat{v} := \frac{M^{-\top}\mathbf{1}}{\sqrt{\mathbf{1}^\top M^{-1} M^{-\top}\mathbf{1}}}$. This finishes the proof for $d = k$.

For $d > k$, the above argument still applies and $H$ remains the only indeterminacy (up to permutation), where $H := I_d - 2\hat{v}\hat{v}^\top$ for $\hat{v} := \frac{(M^\dagger)^\top \mathbf{1}}{\sqrt{\mathbf{1}^\top M^\dagger (M^\dagger)^\top \mathbf{1}}}$. The reason is that even though the ambient dimension $d$ is larger, $\{\mu_i - v : i \in [k]\}$ has to have the same span as $\{\mu_i : i \in [k]\}$, since having the same predictor requires the column space of $M, \tilde{M}$ to match. Hence we only need to consider $v$ in the $k$-dimensional column space of $M$, which reduces to the case of $d = k$.

$\qquad\square$

## A.2  Identifiability of predicting $x_{t_2} \otimes x_{t_3} | x_{t_1}$, G-HMM

Theorem 5 shows that triplet prediction tasks (i.e. predict 2 tokens given 1) suffices for the identifiability of HMM, using tools from the uniqueness of tensor decomposition. The next theorem shows that the same conclusion also applies for G-HMM:

**Theorem 6** (Identifiability from masked prediction on three tokens, G-HMM)**.** *Let $(t_1, t_2, t_3)$ be any permutation of $(1, 2, 3)$, and consider the prediction task $x_{t_2} \otimes x_{t_3} | x_{t_1}$. Under Assumption 1, 4, 5, if the optimal predictors under the G-HMM distributions with parameters $(M, T)$ and $(\tilde{M}, \tilde{T})$ are the same, then $(M, T) = (\tilde{M}, \tilde{T})$ up to a permutation of the hidden state labels.*

*Proof.* Similar to the discrete case, we will prove $x_2 \otimes x_3|x_1$ and $x_1 \otimes x_3|x_2$ separately; the proof for $x_1 \otimes x_2|x_3$ is analogous to $x_2 \otimes x_3|x_1$ by symmetry and hence omitted. The proofs also follow a similar strategy as in the proof for Theorem 5, that is, to construct a 3-tensor using the predictor, on which applying Kruskal's theorem provides identifiability.

**Case 1, $x_2 \otimes x_3|x_1$:** Let $\mathcal{X} := \{x^{(i)} \in \mathbb{R}^d : i \in [k]\}$ be a linearly independent set, and consider the following 3-tensor:

$$
\begin{aligned}
W &:= \sum_{x_i \in \mathcal{X}} x_1 \otimes \mathbb{E}[x_2 \otimes x_3|x_1] = \sum_{x_1 \in \mathcal{X}} x_1 \otimes \mathbb{E}_{h_2|x_1}\big[\mathbb{E}[x_2 \otimes x_3|x_1]|h_2\big] \\
&= \sum_{x_1 \in \mathcal{X}} x_1 \otimes \sum_{h_2} P(h_2|x_1)\mathbb{E}[x_2|h_2] \otimes \mathbb{E}[x_3|h_2] \\
&= \sum_{i \in [k]} \sum_{x_1 \in \mathcal{X}} P(h_2 = i|x_1)x_1 \otimes \mathbb{E}[x_2|h_2 = i] \otimes \mathbb{E}[x_3|h_2 = i] \\
&= \sum_{i \in [k]} \Big( \underbrace{\sum_{x_1} (T\phi(x_1))^\top e_i^{(k)} x_1}_{:=a_i} \Big) \otimes M_i \otimes (MT)_i.
\end{aligned}
\tag{21}
$$

The matrices formed by second and third components are both of rank-$k$ by Assumption 4. Hence in order to apply Kruskal's theorem on $W$, it suffices to show that there exists a choice of $\mathcal{X}$ such that the matrix $A := [a_1, ..., a_k]$ is of rank $k$. One such choice is to let $x^{(i)} = \mu_i$, which gives

$$
\begin{aligned}
a_i &:= \sum_{j \in [k]} \phi(x_1 = \mu_j)^\top T^\top e_i^{(k)} \mu_j = M[\phi(\mu_1), ..., \phi(\mu_k)]^\top T^\top e_i^{(k)}, \\
A &:= [a_1, ..., a_k] = M[\phi(\mu_1), ..., \phi(\mu_k)]^\top T^\top.
\end{aligned}
\tag{22}
$$

Since $M, T$ are both of rank $k$ by Assumption 4, we only need to argue that the matrix $\Phi := [\phi(\mu_1), ..., \phi(\mu_k)] \in \mathbb{R}^{k \times k}$ is of full rank. Recall that for a mixture of $k$ Gaussians with identify covariance and mean $\{\mu_i \in \mathbb{R}^d : i \in [k]\}$, the posterior function $\phi$ is defined entrywise as

$$
[\phi(x)]_i = \frac{\exp\big(-\frac{\|x-\mu_i\|_2^2}{2}\big)}{\sum_{j \in [k]} \exp\big(-\frac{\|x-\mu_j\|_2^2}{2}\big)}, \; \forall i \in [k].
\tag{23}
$$

To show $\Phi$ is of full rank, we can equivalently show that a columnwise scaled version of $\Phi$ is full rank. In particular, let's look at the matrix $\hat{\Phi} \in \mathbb{R}^{k \times k}$, where $\hat{\Phi}_{ij} = \exp(-\frac{\|\mu_i-\mu_j\|^2}{2})$; that is, each column of $\hat{\Phi}$ can be considered as a scaled version of the column in $\Phi$ without the normalization for a unit $\ell_1$ norm. It can be seen that $\hat{\Phi}$ is a Gaussian kernel matrix which is known to be full rank.

Therefore we have shown that each component of the tensor $W := \sum_{i \in [k]} a_i \otimes M_i \otimes (MT)_i$ has Kruskal rank $k$, which allows to recover columns of $M, MT$ up to permutation and scaling by Kruskal's theorem. The indeterminacy in scaling is further removed since the norms of $\{M_i\}_{i \in [d]}$ are known by Assumption 5.

On the other hand, for any $\tilde{M}, \tilde{T}$ that form the same predictor as $M, T, W$ can also be written as

$$
\begin{aligned}
W &= \sum_{x_1 \in \mathcal{X}} x_1 \otimes \mathbb{E}[x_2 \otimes x_3|x_1] = \sum_{x_1 \in \mathcal{X}} x_1 \otimes \tilde{\mathbb{E}}[x_2 \otimes x_3|x_1] \\
&= \sum_{i \in [k]} \Big( \sum_{x_1} (\tilde{T}\tilde{\phi}(x_1))^\top e_i^{(k)} x_1 \Big) \otimes \tilde{M}_i \otimes (\tilde{M}\tilde{T})_i.
\end{aligned}
\tag{24}
$$

Hence columns of $M, \tilde{M}$ and $MT, \tilde{M}\tilde{T}$ are both matched up to a shared permutation, which proves identifiability.

**Case 2, $\mathbb{E}[x_1 \otimes x_3|x_2]$:** For the task of predicting $x_1, x_3$ given $x_2$, the predictor takes the form

$$
\mathbb{E}[x_1 \otimes x_3|x_2] = (OT^\top)\text{diag}(\phi(x_2))(OT)^\top.
\tag{25}
$$

Let $\mathcal{X} := \{\mu_i : i \in [k]\}$ as in the previous case, and consider the 3-tensor

$$
\begin{aligned}
W &:= \sum_{x_2 \in \mathcal{X}} x_2 \otimes \mathbb{E}[x_1 \otimes x_3 | x_2] = \sum_{x_2 \in \mathcal{X}} x_2 \otimes \mathbb{E}_{h_2|x_2}(\mathbb{E}[x_1|h_2] \otimes \mathbb{E}[x_3|h_2]) \\
&= \sum_{h_2} \sum_{x_2 \in \mathcal{X}} p(h_2|x_2) x_2 \otimes \mathbb{E}[x_1|h_2] \otimes \mathbb{E}[x_3|h_2] \\
&= \sum_{i \in [k]} \Big( \underbrace{\sum_{x_2 \in \mathcal{X}} (\phi(x_2))^\top e_i^{(k)} x_2}_{:=a_i} \Big) \otimes (MT^\top)_i \otimes (MT)_i,
\end{aligned}
\tag{26}
$$

The first component is of rank-$k$ as shown in the proof for $x_2 \otimes x_3 | x_1$, and the other two components are of rank-$k$ by Assumption 4. Thus Kruskal's theorem applies and the columns of $MT, MT^\top$ are recovered up to a shared permutation.

The first component $\{a_i\}_{i \in [k]}$ are also recovered, which means that if $\tilde{M}, \tilde{T}$ form the same predictor as $M, T$, then for any linearly independent set $\mathcal{X}$ with $k$ elements (not necessarily the previous choice of $\{\mu_i\}_{i \in [k]}$) such that $\mathcal{X}$ leads to a full rank $A$, we have $A = \tilde{A}$ where $\tilde{A}$ is parameterized by $\tilde{M}, \tilde{T}$. For any such $\mathcal{X} = \{x^{(i)} : i \in [k]\}$, denote $X := [x^{(1)}, ..., x^{(k)}]$, then

$$
A = X[\phi(x^{(1)}), ..., \phi(x^{(k)})]^\top T^\top = X[\tilde{\phi}(x^{(1)}), ..., \tilde{\phi}(x^{(k)})]^\top \tilde{T}^\top = \tilde{A}.
\tag{27}
$$

Since $X$ is of rank-$k$ by the choice of $\mathcal{X}$, this means

$$
[\tilde{\phi}(x^{(1)}), ..., \tilde{\phi}(x^{(k)})] = \underbrace{\tilde{T}^{-1} T}_{:=R} [\phi(x^{(1)}), ..., \phi(x^{(k)})] \Rightarrow \tilde{\phi}(x^{(i)}) = R\phi(x^{(i)}), \ \forall i \in [k].
\tag{28}
$$

Moreover, for any valid choice of $\mathcal{X}$, matrices defined with points in sufficiently small neighborhoods of $x^{(i)}$ are still of full rank by the upper continuity of matrix rank. Hence the equality in equation 28 holds for points in these neighborhoods, and thus the Jacobian on both sides should be equal. Then, the exact same proof of Lemma 2 applies, and we get $\tilde{\phi}, \phi$ are equal up to a permutation of coordinates. Thus $\tilde{M}$ must be equal to (up to permutation) either $M$ or $HM$ for a Householder reflection $H$ by Lemma 1. Finally, the solution of $HM$ is eliminated since it would lead to a $\tilde{T}$ that is not a valid stochastic matrix, as shown in the proof of Theorem 3.

$\square$

## A.3 Identifiability from pairwise conditional distribution

We show that matching the entire conditional *distribution* for G-HMM provides identifiability. Though this is implied by Theorem 3, which states that matching the conditional *expectation* already suffices, having access to the full conditional distribution allows an even simpler proof.

**Theorem 7** (Identifiability of conditional distribution). *Let $M, T$ and $\tilde{M}, \tilde{T}$ be two set of parameters satisfying Assumption 1 and 4. If $p(x_2|x_1; M, T) = p(x_2|x_1; \tilde{M}, \tilde{T})$, $\forall x_1, x_2 \in \mathbb{R}^d$, then $M = \tilde{M}$, $T = \tilde{T}$ up to a permutation of labeling.*

*Proof.* First note that the conditional distribution of $x_2$ given $x_1$ is a mixture of Gaussian, with means $\{\mu_i\}_{i \in [k]}$ and mixture weights given by $P(h_2|x_1) = TP(h_1|x_1)$, hence we can directly apply the identifiability of Gaussian mixtures to recover the means $\{\mu_i\}_{i \in [k]}$:

**Lemma 3** (Proposition 4.3 in Lindsay and Basak [1993]). *Let $Q_k$ denote a Gaussian mixture with means $\{\xi_j\}_{j \in [k]} \in \mathbb{R}^d$. Suppose $\exists l \in [d]$ such that the set $\{[\xi_j]_l\}$ has distinct values, then one can recover $\{\xi_j\}_{j \in [k]}$ from moments of $Q_k$.*

We note that the assumption on the existence of a coordinate $l \in [k]$ is with out loss of generality, since we can first rotate the means to a different coordinate system in which this condition holds, then rotation back the means. Such rotation is guaranteed to exist since finding such rotation is equivalent to finding a vector $v$ s.t. $v^\top(\mu_i - \mu_j) \neq 0$ for every $i, j \in [k]$, for which the solution set is $\mathbb{R}^d \setminus \cup_{i,j \in [k]} \{u : u^\top(\mu_i - \mu_j) = 0\} \neq \emptyset$.

Recovering $\{\mu_i\}_{i\in[k]}$ means the scaled likelihood and the posterior both match, i.e. $\psi = \tilde{\psi}$, and $\phi(x) = P(h|x) = \frac{\psi}{\|\psi\|_1}$. The conditional distribution is

$$p(x_2|x_1) = \sum_{i,j\in[k]} p(x_2|h_2)p(h_2|h_1)p(h_1|x_1) = \frac{1}{(2\pi)^{d/2}}\psi(x_2)^\top T\phi(x_1). \tag{29}$$

Choose a set $\mathcal{X} := \{x^{(i)}\}_{i\in[k]}$ such that $\Psi_\mathcal{X} := [\psi(x^{(1)}), ..., \psi(x^{(k)})] \in \mathbb{R}^{k\times k}$ is full rank. $\Phi_\mathcal{X} := [\phi(x^{(1)}), ..., \phi(x^{(k)})] \in \mathbb{R}^{k\times k}$ is also full rank since its columns are nonzero scalings of columns of $\Psi_\mathcal{X}$. Then we have

$$\Psi_\mathcal{X}^\top T\Phi_\mathcal{X} = \tilde{\Psi}_\mathcal{X}^\top \tilde{T}\tilde{\Phi}_\mathcal{X} = \Psi_\mathcal{X}^\top \tilde{T}\Phi_\mathcal{X} \Rightarrow T = \tilde{T}. \tag{30}$$

$\square$

# B  Proof of Theorem 4: non-identifiability of HMM from multiple pairwise predictions

**Theorem** (Theorem 4 restated: nonidentifiability of HMM from multiple pairwise predictions). *There exists a pair of HMM distributions with parameters $(O, T)$ and $(\tilde{O}, \tilde{T})$, each satisfying Assumptions 1, 2 and 3, and also $\tilde{O} \neq O$, such that, for each of the tasks $x_2|x_1$, $x_1|x_2$, $x_3|x_1$, and $x_1|x_3$, the optimal predictors are the same under each distribution.*

*Proof.* We provide an example to show the nonidentifiability result in Theorem 4. The goal is to find $\tilde{O} \neq O$, $\tilde{T} \neq T$ that produce the same predictors for predicting both $x_2|x_1$ and $x_3|x_1$. We will choose $T, \tilde{T}$ to be symmetric, so that $O, T$ and $\tilde{O}, \tilde{T}$ also form the same predictors for the reversed direction, i.e., for predicting $x_1$ given $x_2$ and $x_1$ given $x_3$, since the reverse chain has transition matrix $T^\top = T$.

Let's consider the case where the all row sums of $O$ and $\tilde{O}$ are $k/d$. Consequently, the posterior function is simply $\phi(x) = \frac{O^\top x}{\|O^\top x\|_1} = \frac{d}{k}O^\top x$, and similarly we have $\tilde{\phi}(x) = \frac{d}{k}\tilde{O}^\top x$. The predictors are of the form:

$$f^{2|1}(x) = OT\phi(x) = \frac{d}{k}OTO^\top x, \quad f^{3|1}(x) = OT^2\phi(x) = \frac{d}{k}OT^2O^\top x. \tag{31}$$

Matching $f^{2|1}(x) = \tilde{f}^{2|1}(x)$ on all $x \in \mathcal{X} := \{e_i\}_{i\in[d]}$ means

$$OTO^\top I_d = OTO^\top = \tilde{O}\tilde{T}\tilde{O}^\top \Rightarrow \tilde{T} = \tilde{O}^\dagger O \cdot T \cdot (\tilde{O}^\dagger O)^\top. \tag{32}$$

Similarly, matching $f^{3|1} = \tilde{f}^{3|1}$ gives $OT^2O^\top = \tilde{O}\tilde{T}^2\tilde{O}^\top$, hence

$$\tilde{O}\tilde{T}^2\tilde{O}^\top = \tilde{O}\tilde{O}^\dagger OT(\tilde{O}^\dagger O)^\top \cdot \tilde{O}^\dagger OT(\tilde{O}^\dagger O)^\top \tilde{O}^\top$$

$$\stackrel{(i)}{=} OT \cdot (\tilde{O}^\dagger O)^\top \tilde{O}^\dagger O \cdot TO^\top = OT \cdot TO^\top \Rightarrow (\tilde{O}^\dagger O)^\top \cdot \tilde{O}^\dagger O = I_k, \tag{33}$$

where step $(i)$ uses $\tilde{O}\tilde{O}^\dagger O = O$, since $\tilde{O}, O$ share the same column space.

Denote $R := \tilde{O}^\dagger O$; $R$ is orthogonal by the last equality in equation 33. To construct the desired example, consider $k = 3$, and let $R$ represent a rotation with axis of rotation $\frac{1}{3}(e_1 + e_2 + e_3)$. This axis is the direction pointing from the origin to the projection of the origin on the hyperplane $\mathcal{P}_c := \{v \in \mathbb{R}^d : \sum_{i\in[d]} v_i = c\}$ for any positive constant $c$ (i.e. $\mathcal{P}_c$ is parallel to the hyperplane in which probability vectors lie). This means such rotation guarantees $Rv \in \mathcal{P}_c, \forall v \in \mathcal{P}_c$, and has the following property:

**Claim 4.** *Each row and each column of $R$ sums up to 1.*

Define $\tilde{O} := OR$, $\tilde{T} := R^\top TR$, Claim 4 ensures that row sum and column sum of $\tilde{O}, \tilde{T}$ remain the same as those of $O, T$. When the rotation angle represented by $R$ is sufficiently small, entries $\tilde{O}, \tilde{T}$ remain in $[0, 1]$, hence such $\tilde{O}, \tilde{T}$ form a valid example. We will provide a concrete example in the subsequent subsection.

$\square$

## B.1 Example for Theorem 4

The intuition of the nonidentifiability result in Theorem 4 is related to the non-uniqueness of matrix factorization: while adding additional pairwise prediction tasks introduces more equations on the product of matrices, these equations can be highly dependent, and there are cases where different set of matrices can simultaneously satisfy all the equations.

We now provide a concrete example for the non-identifiability of predicting $x_2|x_1$, $x_1|x_2$, $x_3|x_1$, and $x_1|x_3$, by finding 2 set of $O, T$ such that the corresponding predictors (of the form specified in equation 31) match. Let $d = 4$, $k = 3$,

$$O = \begin{bmatrix} 0.23016003 & 0.3549092 & 0.16493077 \\ 0.30716059 & 0.06962305 & 0.37321636 \\ 0.2580854 & 0.26965425 & 0.22226035 \\ 0.20459398 & 0.3058135 & 0.23959252 \end{bmatrix}, \tilde{O} = \begin{bmatrix} 0.24120928 & 0.35062535 & 0.15816537 \\ 0.28937626 & 0.07433156 & 0.38629218 \\ 0.26077674 & 0.26749114 & 0.22173212 \\ 0.20863772 & 0.30755194 & 0.23381033 \end{bmatrix},$$

$$T = \begin{bmatrix} 0.56893146 & 0.35811118 & 0.07295736 \\ 0.35811118 & 0.10805638 & 0.53383243 \\ 0.07295736 & 0.53383243 & 0.39321021 \end{bmatrix}, \tilde{T} = \begin{bmatrix} 0.59740926 & 0.30452087 & 0.09806987 \\ 0.30452087 & 0.1331689 & 0.56231024 \\ 0.09806987 & 0.56231024 & 0.33961989 \end{bmatrix},$$

$$\det(O) = \det(\tilde{O}) = 0.0110, \det(T) = \det(\tilde{T}) = -0.1611.$$
(34)

Note that $T, \tilde{T}$ are both symmetric as desired by the proof of Theorem 4, which means this is also a valid counter example for learning to predict $x_1|x_2$ and $x_1|x_3$, and hence for all of $x_2|x_1$, $x_1|x_2$, $x_3|x_1$, and $x_1|x_3$.

## B.2 Proof of Claim 4

*Proof.* We would like to show that each row and each column of $R$ sums up to 1. Denote the $d$-dimensional simplex by $\Delta_d$, i.e. $\Delta_d := \{x \in \mathbb{R}^d : \sum_{i \in [d]} x_i = 1\}$, and let $\mathcal{P}_c := \{v \in \mathbb{R}^d : \sum_{i \in [d]} v_i = c\}$ for some positive constant $c$ denote a hyperplane parallel to the hyperplane in which probability vectors lie.

Let's first check that the columns of $R$ sum up to 1. Any $v \in \mathcal{P}_c$ can be written as $v = c \cdot [\alpha_1, \alpha_2, ..., \alpha_{d-1}, 1 - \sum_{i \in [d-1]} \alpha_i]$ for some $[\alpha_1, ..., \alpha_{d-1}] \in \Delta_{d-1}$. Let $r_i$ denote the $i_{th}$ row of $R$, then $Rv \in \mathcal{P}_c$ means $\sum_{i \in [d]} \langle r_i, v \rangle = \langle \sum_{i \in [d]} r_i, v \rangle = c$. Let $\beta_j$ denote the $j_{th}$ coordinate of $\sum_{i \in [d]} r_i$, then

$$\sum_{i \in [d-1]} \beta_i \alpha_i + \beta_d \left(1 - \sum_{i \in [d-1]} \alpha_i\right) = 1, \forall [\alpha_1, ..., \alpha_{d-1}] \in \Delta_{d-1}$$

$$\Rightarrow \sum_{i \in [d-1]} (\beta_i - \beta_d)\alpha_i + \beta_d = 1, \forall [\alpha_1, ..., \alpha_{d-1}] \in \Delta_{d-1}$$
(35)

$$\Rightarrow \beta_i = 1, \forall i \in [d].$$

It then follows that $R^{-1} = R^\top$ also has columns summing up to 1, since

$$\sum_{i \in [d]} (RR^{-1})_{ij} = \langle \sum_{i \in [d]} r_i, (R^{-1})_j \rangle = \langle \mathbf{1}, (R^{-1})_j \rangle = 1, \forall j \in [d].$$
(36)

$\square$

## C Nonidentifiability from large time gaps

As noted earlier, there is an inherent obstacle when using prediction tasks on tokens that are more than 1 time gaps apart. For instance, if we are predicting $x_{t+1}$ given $x_1$ for some $t > 1$ with G-HMM, then we are still able to identify $M$ from the posterior function, however it remains to to recover $T$ from $T^t$. For general matrices, it is clear that matching a power of a matrix does not imply the matrix itself is matched. For our case, even though requiring $T$ to be stochastic adds additional constraints,

matching the matrix power still does not suffice to identify the underlying matrix, as formalized in the following claim.

**Claim 5** (Nonidentifiability of matrix powers (Claim 1 restated)). *For any positive integer $t$, there exist stochastic matrices $T, \tilde{T}$ satisfying Assumption 1, 3, such that $T \neq \tilde{T}$ and $T^t = \tilde{T}^t$.*

*Proof.* As in Theorem 4, the nonidentifiability comes from the non-uniqueness of matrix factorization. Specifically for this case, we will set $\tilde{T}$ to be equal to $T$ up to a special rotation that gets composed when taking the matrix power. That is, we want $\tilde{T} = RT = TR$ for some matrix $R$ that implicitly performs a rotation, so that $\tilde{T}^t = T^t R^t$. Since $R$ corresonds to a rotation, we can choose the rotation angle properly so that $R^t = I$, and hence $\tilde{T}^t = T^t$ but $\tilde{T} \neq T$.

Precisely, using notations for the G-HMM setup, set $a \in [0, 1]$, and let the parameters $(T, M)$ be given by

$$T = \begin{bmatrix} a & 0 & 1-a \\ 1-a & a & 0 \\ 0 & 1-a & a \end{bmatrix}, \quad M = \begin{bmatrix} 1 & -1/2 & -1/2 \\ 0 & -\sqrt{3}/2 & \sqrt{3}/2 \\ 1/\sqrt{2} & 1/\sqrt{2} & 1/\sqrt{2} \end{bmatrix}.$$

Let $\theta$ be some rotation angle, and denote by $R(\theta) := \begin{bmatrix} \cos(\theta) & -\sin(\theta) & 0 \\ \sin(\theta) & \cos(\theta) & 0 \\ 0 & 0 & 1 \end{bmatrix}$ a rotation that acts on the first two dimensions. We will show that for any $\theta \in \mathbb{R}$, we have

$$\tilde{T} := \left( M^{-1} \left( R(\theta) \right)^{-1} M \right) \cdot T = T \cdot \left( M^{-1} \left( R(\theta) \right)^{-1} M \right). \tag{37}$$

Assuming equation 37, since $R(\theta)$ represents a rotation of angle $\theta$, $\left( R(\theta) \right)^{\tau}$ corresponds to a rotation of angle $\tau\theta$ for any integer $\tau$ ($\tau$ could be negative). Setting $\theta := \frac{2\pi}{t}$, we then have

$$\tilde{T}^t = \left( M^{-1} \left( R(\theta) \right)^{-1} M \cdot T \right)^t = T^t \left( M^{-1} \left( R(\theta) \right)^{-1} M \right)^t = T^t M^{-1} \left( R(\theta) \right)^{-t} M$$
$$= T^t M^{-1} \cdot R(2\pi) \cdot M = T^t. \tag{38}$$

For $\tilde{T}$ to serve as a valid example for our theorem, it remains to check that for every $t$, there exists a choice of $a$ such that $\tilde{T} := RT$, where $R := M^{-1} \left( R(\frac{2\pi}{t}) \right)^{-1} M$, is a valid stochastic matrix. That is, $\tilde{T}$ has 1) columns and rows each summing up to 1, and 2) entries bounded in $[0, 1]$. Let's first show that the columns and rows each sum up to 1. Noting that $M^{-1} = \frac{1}{3} \begin{bmatrix} 2 & 0 & \sqrt{2} \\ -1 & -\sqrt{3} & \sqrt{2} \\ -1 & \sqrt{3} & \sqrt{2} \end{bmatrix}$, the column sums are

$$\mathbf{1}^\top \tilde{T} = \mathbf{1}^\top M^{-1} R(\theta)^{-1} M T \overset{(i)}{=} \mathbf{1}^\top T M^{-1} R(\theta)^{-1} M = \sqrt{2} e_3^\top R(\theta) M = \sqrt{2} e_3^\top M = \sqrt{2} \frac{1}{\sqrt{2}} \mathbf{1} = \mathbf{1}, \tag{39}$$

where step $(i)$ uses equation 37. Similarly, the row sums are

$$\tilde{T}\mathbf{1} = M^{-1} R(\theta)^{-1} M \mathbf{1} = M^{-1} R(\theta)^{-1} \cdot \frac{3}{\sqrt{2}} e_3 = M^{-1} \cdot \frac{3}{\sqrt{2}} e_3 = \mathbf{1}. \tag{40}$$

To show that there exists a choice of $T$ such that entries of $\tilde{T}$ are non-negative, we provide a concrete example where $T$ is defined with $a = \frac{1}{2}$. It can be checked that $\tilde{T} := M^{-1}(R(\frac{2\pi}{t}))^{-1}M$ has non-negative entries for $t \in \{2, 3, 4, ..., 10\}$. For larger $t$, let $\theta = \frac{2\pi}{t}$, then we have by the Taylor expansion of $R(\frac{2\pi}{t})$:

$$R(\theta) := \begin{bmatrix} \cos\theta & -\sin\theta & 0 \\ \sin\theta & \cos\theta & 0 \\ 0 & 0 & 1 \end{bmatrix} = \begin{bmatrix} 1 - \theta^2/2 + c_1\theta^4 & -\theta + c_2\theta^2 & 0 \\ \theta + c_2\theta^2 & 1 - \theta^2/2 + c_1\theta^4 & 0 \\ 0 & 0 & 1 \end{bmatrix}$$
$$= I + \theta \begin{bmatrix} 0 & -1 & 0 \\ 1 & 0 & 0 \\ 0 & 0 & 0 \end{bmatrix} + \theta^2 \begin{bmatrix} -1/2 + c_1\theta^2 & c_2 & 0 \\ c_2 & -1/2 + c_1\theta^2 & 0 \\ 0 & 0 & 0 \end{bmatrix} \tag{41}$$

for some constants $c_1 \in [-\frac{1}{4!}, \frac{1}{4!}]$, $c_2 \in [-\frac{1}{2}, \frac{1}{2}]$. Substituting this into $\tilde{T} := M^{-1}R(\theta)^{-1}M$ gives

$$
\tilde{T} = \begin{bmatrix} a - \frac{\theta}{\sqrt{3}}(1-a) & \frac{\theta}{\sqrt{3}}(1-2a) & 1-a+\frac{\theta}{\sqrt{3}}a \\ 1-a+\frac{\theta}{\sqrt{3}}a & a - \frac{\theta}{\sqrt{3}}(1-a) & \frac{\theta}{\sqrt{3}}(1-2a) \\ \frac{\theta}{\sqrt{3}}(1-2a) & 1-a+\frac{\theta}{\sqrt{3}}a & a-\frac{\theta}{\sqrt{3}}(1-a) \end{bmatrix}
$$

$$
+ \frac{1}{3}\begin{bmatrix} -1+2c_1\theta^2 & \frac{1}{2}-c_1\theta^2-\sqrt{3}c_2 & \frac{1}{2}-c_1\theta^2+\sqrt{3}c_2 \\ \frac{1}{2}-c_1\theta^2-\sqrt{3}c_2 & -1+2c_1\theta^2+\sqrt{3}c_2 & \frac{1}{2}-c_1\theta^2 \\ \frac{1}{2}-c_1\theta^2+\sqrt{3}c_2 & \frac{1}{2}-c_1\theta^2 & -1+2c_1\theta^2-\sqrt{3}c_2 \end{bmatrix} \cdot \begin{bmatrix} a & 0 & 1-a \\ 1-a & a & 0 \\ 0 & 1-a & a \end{bmatrix}
$$

$$
= \frac{1}{2}\begin{bmatrix} 1-\frac{\theta}{\sqrt{3}} & 0 & 1+\frac{\theta}{\sqrt{3}} \\ 1+\frac{\theta}{\sqrt{3}} & 1-\frac{\theta}{\sqrt{3}} & 0 \\ 0 & 1+\frac{\theta}{\sqrt{3}} & 1-\frac{\theta}{\sqrt{3}} \end{bmatrix} + \frac{\theta^2}{6}\begin{bmatrix} -\frac{1}{2}+c_1\theta^2-\sqrt{3}c_2 & 1-2c_1\theta^2 & -\frac{1}{2}+c_1\theta^2+\sqrt{3}c_2 \\ -\frac{1}{2}+c_1\theta^2 & -\frac{1}{2}+c_1\theta^2+\sqrt{3}c_2 & 1-2c_1\theta^2-\sqrt{3}c_2 \\ 1-2c_1\theta^2+\sqrt{3}c_2 & -\frac{1}{2}+c_1\theta^2-\sqrt{3}c_2 & -\frac{1}{2}+c_1\theta^2 \end{bmatrix}
$$

$$
\overset{(i)}{\geq} \frac{1}{2}\begin{bmatrix} 1-\frac{\theta}{\sqrt{3}} & 0 & 1+\frac{\theta}{\sqrt{3}} \\ 1+\frac{\theta}{\sqrt{3}} & 1-\frac{\theta}{\sqrt{3}} & 0 \\ 0 & 1+\frac{\theta}{\sqrt{3}} & 1-\frac{\theta}{\sqrt{3}} \end{bmatrix} + \theta^2\begin{bmatrix} -0.25 & -0.16 & -0.25 \\ -0.09 & -0.25 & 0.01 \\ 0.01 & -0.25 & -0.09 \end{bmatrix}
\tag{42}
$$

where the inequality $(i)$ is taken entry-wise. It can be checked that all entries are non-negative for $\theta \leq \frac{2\pi}{10}$.

**Proof of equation 37**    Let's conclude the proof by proving the commutativity in equation 37. Denote $R_2(\theta) := \begin{bmatrix} \cos(\theta) & -\sin(\theta) \\ \sin(\theta) & \cos(\theta) \end{bmatrix}$, i.e. $R(\theta) = \begin{bmatrix} R_2(\theta) & 0 \\ 0 & 1 \end{bmatrix}$. Denote $U := \begin{bmatrix} 1 & -1/2 & -1/2 \\ 0 & -\sqrt{3}/2 & \sqrt{3}/2 \end{bmatrix}$, i.e. $M = \begin{bmatrix} U \\ \mathbf{1}^\top/\sqrt{2} \end{bmatrix}$. We can write

$$
M^\top R(\theta)^\top M = \begin{bmatrix} U^\top & \mathbf{1}/\sqrt{2} \end{bmatrix}\begin{bmatrix} R_2(\theta)^\top & 0 \\ 0 & 1 \end{bmatrix}\begin{bmatrix} U \\ \mathbf{1}^\top/\sqrt{2} \end{bmatrix} = U^\top R_2(\theta)^\top U + \frac{\mathbf{1}\mathbf{1}^\top}{2}.
\tag{43}
$$

Let $R_2(\theta)$ denote a clockwise rotation of angle $\theta$, then

$$
U = [v_1, R_2(\tfrac{2\pi}{3})v_1, R_2(\tfrac{4\pi}{3})v_1] = [R_2(\tfrac{4\pi}{3})v_2, v_2, R_2(\tfrac{2\pi}{3})v_2] = [R_2(\tfrac{2\pi}{3})v_3, R_2(\tfrac{4\pi}{3})v_3, v_3],
\tag{44}
$$

where $v_1 = \begin{bmatrix} 1 \\ 0 \end{bmatrix}$, $v_2 = \begin{bmatrix} -1/2 \\ -\sqrt{3}/2 \end{bmatrix}$, $v_3 = \begin{bmatrix} -1/2 \\ \sqrt{3}/2 \end{bmatrix}$. Denote $\alpha_{ij} := v_i^\top R_2^\top v_j$ for $i,j \in [3]$. Noting $T = aI + (1-a)\begin{bmatrix} 0 & 0 & 1 \\ 1 & 0 & 0 \\ 0 & 1 & 0 \end{bmatrix} := aI + (1-a)P$, we have

$$
M^\top R(\theta)^\top M T = T M^\top R(\theta)^\top M
$$

$$
\Leftrightarrow U^\top R_2(\theta)^\top U(aI + (1-a)P) + \frac{\mathbf{1}\mathbf{1}^\top}{2}T = (aI + (1-a)P)U^\top R_2(\theta)^\top U + T\frac{\mathbf{1}\mathbf{1}^\top}{2}
$$

$$
\overset{(i)}{\Leftrightarrow} U^\top R_2(\theta)^\top U P = P U^\top R_2(\theta)^\top U
\tag{45}
$$

$$
\Leftrightarrow \begin{bmatrix} \alpha_{31} & \alpha_{32} & \alpha_{33} \\ \alpha_{11} & \alpha_{12} & \alpha_{13} \\ \alpha_{21} & \alpha_{22} & \alpha_{23} \end{bmatrix} \overset{(*)}{=} \begin{bmatrix} \alpha_{12} & \alpha_{13} & \alpha_{11} \\ \alpha_{22} & \alpha_{23} & \alpha_{21} \\ \alpha_{32} & \alpha_{33} & \alpha_{31} \end{bmatrix}.
$$

where step $(i)$ uses $\mathbf{1}\mathbf{1}^\top T = T\mathbf{1}\mathbf{1}^\top = \mathbf{1}\mathbf{1}^\top$. The equality $(*)$ is true due to equation 44.     $\square$

## D    Simulation

We empirically verify the identifiability results for HMM (Theorem 5) and G-HMM (Theorem 3) on simulation data, by checking whether matching the optimal predictor implies matching the parameters of the data generative model.

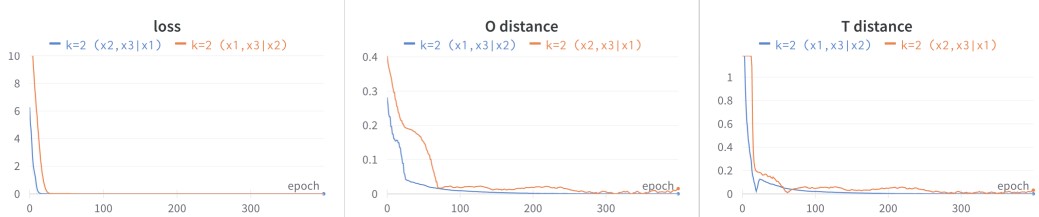

Figure 1: HMM: left: objective; middle: $\|O - O^*\|_F$; right: $\|T - T^*\|_F$.

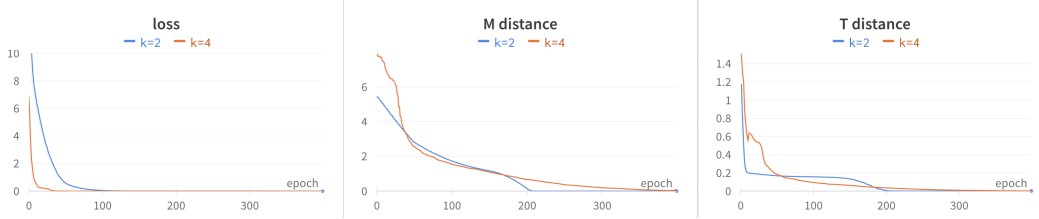

Figure 2: G-HMM: left: objective; middle: $\|M - M^*\|_F$; right: $\|T - T^*\|_F$.

In particular, we aim to recover $O \in \mathbb{R}^{d \times k}, T \in \mathbb{R}^{k \times k}$ for HMM, and $M \in \mathbb{R}^{d \times k}, T \in \mathbb{R}^{k \times k}$ for G-HMM, with $d = 10$ and $k \in \{2, 4\}$. Given a batch of samples $\mathcal{B} := \{x_1^{(i)}\}_{i \in |\mathcal{B}|}$, the objective to minimize for HMM is

$$\ell(O, T) := \frac{1}{|\mathcal{B}|} \sum_{x_1 \in \mathcal{B}} \|f_{O^*, T^*}^{2,3|1}(x_1) - f_{O,T}^{2,3|1}(x_1)\|_F^2, \tag{46}$$

where $f_{O,T}^{2,3|1}(x_1) := \mathbb{E}_{O,T}[x_2 \otimes x_3 | x_1]$; the objective for the task of predicting $x_1 \otimes x_3$ given $x_2$ is defined analogously. Note that though equation 46 differs from equation 1 by a constant, [8] it suffices for verifying parameter identifiability since both losses are minimized at $f_{O,T} = f_{O^*, T^*}$. We choose to use the form in equation 46 since it is more stable to optimize for and that its minimal loss value is 0, making it easy to check for optimality. Similarly, the objective to minimizer for G-HMM is

$$\ell(M, T) := \frac{1}{|\mathcal{B}|} \sum_{x_1 \in \mathcal{B}} \|f_{M^*, T^*}^{2|1}(x_1) - f_{M,T}^{2|1}(x_1)\|_2^2, \tag{47}$$

where $f_{M,T}^{2|1} := \mathbb{E}_{M,T}[x_2 | x_1]$.

For both HMM and G-HMM, we optimize for $O$ (or $M$) and $T$ alternatingly in different epochs. We found that it is usually helpful to use a larger learning rate for $T$ than for $O$ (or $M$), and that normalized gradient descent helps speed up training [9]

Figure 1 and 2 show the results for HMM and G-HMM. It can be seen that as the objective value approaches the optimum, the parameter distances indeed go to zero, corroborating Theorem 5 and 3.

---

[8]In equation 1, the population loss is defined as $\mathbb{E}_{x_1} \mathbb{E}_{x_2, x_3 | x_1} \|x_2 \otimes x_3 - f(x_1)\|_F^2$, whereas for equation 46 the population loss is $\mathbb{E}_{x_1} \|\mathbb{E}_{x_2, x_3 | x_1}[x_2 \otimes x_3] - f(x_1)\|_F^2$, i.e. the expectation over $x_2, x_3$ is moved to within the Frobenius norm. The two losses differ by a constant that is independent of the parameters of $f$.

[9]That is, we normalize each gradient to have Frobenius norm 1. The gradients are otherwise too small which will result in slow convergence.