# OpenReview forum: "Masked Prediction: A Parameter Identifiability View"
_NeurIPS.cc/2022/Conference — NeurIPS 2022 Accept_

### Official Review · Reviewer_TcPc · 2022-07-11

**Rating:** 7
**Confidence:** 3
**Soundness:** 4 excellent
**Presentation:** 3 good
**Contribution:** 3 good

**Summary:**

The authors propose a novel theoretical way of analyzing self supervised learning by model identifiability. HMM and Gaussian HMM models are assumed for latent space sequences. The proved theorems provide 3 main claims: non-identifiability in pair-wise prediction under HHM model on 2 and 3 tokens, identifiability of pair-wise predictions under G-HHM model, identifiability of masked pair-wise predictions on 3 tokens under HMM.

**Questions:**

I think it a typo in line 117: O_{x,j} should be O_{j, x} according to the definition in line 96


**Limitations:**

The limitation are discussed properly both in conclusion and paper body (e.g. Assumption 5)

**Strengths And Weaknesses:**

Strengths:

- First effort of analyzing self supervised learning by model identifiability

- Formulated and proved the theorems that could be a foundation of the identifiability analysis of SSL

Weaknesses:
- It could be easier to follow the analysis if some additional examples were added that

---

> ### Author Response · Authors · 2022-07-30
> **Thank you for the encouraging feedback and suggestions**
>
> We thank the reviewer for the careful read and the encouraging feedback!
>
> Please note that we have fixed the typos in the revision, and added some high-level explanations and intuitions for proofs in the appendix (changes highlighted in red) to make them easier to follow.
>
> We’d also like to mention that there seems to be an incomplete sentence (possibly a typo) in the comment on weaknesses; please let us know if there are any other concerns, thank you!

---

### Official Review · Reviewer_RbEr · 2022-07-11

**Rating:** 6
**Confidence:** 3
**Soundness:** 3 good
**Presentation:** 3 good
**Contribution:** 3 good

**Summary:**


This paper investigates the identifiability of latent stochastic generative parameters in the context of masking self-supervised prediction tasks. This is a theoretical contribution; this work does not provide experimental results on how well these findings hold in practice when training masking prediction tasks but shows under what conditions an optimal predictor allows the identification of latent parameters (experimental simulations in the appendix).

This paper looks at identifiability guarantees of parametric generative models when training self-supervised learning tasks. In particular, the paper focuses on:
- Hidden Markov models: sequential stochastic generative models of two types a) discrete HMM where both hidden states and observations are discrete b) Gaussian HMM where hidden states are discrete and observations follow a conditional Gaussian distribution. In addition, a set of assumptions (stochasticity of transitions, non-redundancy & non-degeneracy of transitions and emissions, equality of norms of observation means) on the HMMs are defined.
- Masking Prediction tasks: the paper focuses on the self-supervised learning task of using part of a sequence of observations to predict the missing part of the sequence.

The contributions of the paper are the following:
- gives identifiability results for G-HMM
- gives identifiability results for discrete HMM for multi-sequence predictions tasks using the uniqueness of tensor rank decomposition and Kruskal's identifiability theorem.

**Questions:**

Questions related to limitations were added to the limitations section.


Typos:
- line 252
- line 255
- line 206
- line 52
- line 114

**Limitations:**

The authors briefly mention that extending to more complex latent variable models and trying to relax assumption 5 are potential extensions of this work that would address some of the limitations.

The authors focus on the predictive tasks where one input observation/token is used to predict one output observation/token. In practice, masking prediction tasks often consider multiple time steps sequence predictions (multiple time-steps in the input & output). Would this setup be a limiting case for this work? Do the authors have some intuition on whether the identifiability results presented in this paper can be extended to this practical setup?

Disclaimer: I would like to mention that I do not have a theoretical background, I did go through most of the proofs and based on my knowledge tried to assess the soundness of the approach.

**Strengths And Weaknesses:**

Strengths:
- as mentioned by the authors, identifiability allows for reducing the assumptions under which performance of the (self-supervised) method holds, in particular assumptions on the downstream performance. Better understanding under which conditions the underlying generative parameters are identifiable is valuable to the field; This work contributes to answering these questions for the case of masked predictive tasks which are commonly used tasks in the field when working with sequential data.
- linked to the previous comment, this work is well motivated and put into context. The motivation behind why identifiability guarantees are valuable is well stated/written. The contributions in comparison to prior art in the field of self-supervised learning and non-linear ICA are also clearly stated (see comment below regarding completeness of the literature review).
- the parametric models considered in this work are quite simplistic setups but remain relevant/realistic and the majority of the set of assumptions (set aside the equality of norms of means - assumption 5) are mild.

Weaknesses:
- as briefly mentioned below, the considered setup is restricted to simple latent variable models which might not hold in practical settings
- the authors claim that one contribution of this work is to propose an alternative parameter identifiability lens for evaluating self-supervised settings. Whilst the specific models and tasks considered in this paper might be novel, the identifiability of generative models in self-supervised learning setups have been investigated in the past and well-motivated. In the case of contrastive learning & non-linear ICA it is worth mentioning the works of Zimmermann et al. 2020 and von Kügelen et al. 2021 which are not present in the references.

---

> ### Author Response · Authors · 2022-07-30
> **Thank you for the constructive comments & discussions on limitations**
>
> We thank the reviewer for their constructive comments and for the encouraging words ! We are glad you find our work “valuable”, “well-motivated” and “put into context”.  We reply to the concerns below:
>
> * _The setup seems restrictive for practical applications_: We agree with the reviewer that the gap between our toy models and real-world models for complex domains like language and images is still substantial. Nonetheless, as the first work on studying SSL through the lens of parameter identifiability, we view the probabilistic setup in our work as a starting point for theoretical inquiries. For these initial results, we start with simpler models to develop necessary theoretical tools, which we hope to extend to incorporate more realistic assumptions in future works. One potential pathway to connect the theoretical insights to applications is to find plausible structural models for practical time-series data (e.g., natural language texts, or continuous signals such as speech and EEG) which can also be theoretically analyzed. As an example, Zhang & Hashimoto 21 uses Gaussian graphical models to represent linguistic structures (such as dependencies), though the question of finding proper structural models is still open for unsupervised learning at large.
>
>   Reference: Zhang & Hashimoto 21: On the Inductive Bias of Masked Language Modeling: From Statistical to Syntactic Dependencies.
>
> * _Other masking schemes_: This is an interesting question. For the setups we consider, given that predicting 2 tokens given 1 already suffice for identifiability, we conjecture that predicting multiple tokens given multiple would also suffice—though this doesn’t immediately follow from our results and certainly would be interesting to prove! For more realistic setups where more complex latent models are needed, we imagine that more difficult masking tasks might be required. In general, we hope that our work can serve as a set of first-cut results that could benefit such future research.
>
> * _Related works on identifiability results for self-supervised methods_: Thank you for pointing out these two papers! We didn't initially discuss these papers because these papers address identifiability on the latent variables, whereas our work focuses on the identifiability of (the parameters of) the generative model itself, which is a complementary notion of identifiability to latent recovery and has been understudied in prior work. But in retrospect, we agree with the reviewer that discussing these works is warranted to emphasize the previous connection between identifiability in general and SSL. We have included these references in the revised version (around line 352, highlighted in red).
>
> Thank you also for pointing out the typos, we have corrected them in the revision.

---

> > ### Author Response · Authors · 2022-08-08
> > **Thank you again for the review!**
> >
> > Thank you very much again for the helpful feedback! Please let us know if our response has addressed your concerns, and we are happy to discuss more if there are further questions.

---

### Official Review · Reviewer_wW1j · 2022-07-11

**Rating:** 6
**Confidence:** 4
**Soundness:** 4 excellent
**Presentation:** 4 excellent
**Contribution:** 3 good

**Summary:**

The traditional way to assess self-supervised learning algorithms is by designing downstream tasks which, if learning is to be deemed successful, will be solved with high accuracy; this way relies on features being learnt which will be useful to solving the downstream task. In contrast, this work provides a theoretical contribution to the understanding of SSL by assessing performance on the task of identifying the parameters of a realizable model (i.e. a model from a class assumed to contain the data-generating model). It does so on discrete- and Gaussian-valued HMMs, working out formal conditions for the identifiability from different types of training observations, such as predicting the mean of a token conditional on another one.

**Questions:**

Can you outline what is needed to turn this work into insights relevant to practical SSL uses? What is the pathway to impact?
How can the formal setup you use (HMMs, masked prediction tasks) be extended to sequence prediction models, neural seq2seq models?

**Limitations:**

Yes

**Strengths And Weaknesses:**

The paper is very clear, the formalization has no unnecessary complications and the notation is clear and explained well. All arguments and proofs which I could check are correct and relevant. As such the paper provides, within its scope, a definitive answer to those questions it addresses.
The paper is relatively significant: understanding SSL is important because it is used so widely, yet understood so little. This paper is also original, as only little work takes an investigative approach different from "assess SSL on downstream task performance".
On the other hand, the setup under study is narrow and strongly simplified to support a formal investigation. Practical impact seems distant. The paper is limited to a theoretical investigation, which it solves well from an applied maths point of view, but does not venture into experimental work.

---

> ### Author Response · Authors · 2022-07-30
> **Thank you for your review & discussion on practical impact**
>
> We thank the reviewer for their thoughtful comments, and for recognizing the work is “clear” and “original”!
>
> The main concern is about the practical impact of the results. On a high level, our work aims to understand what tasks are sufficient to learn a data model, which is a central question for self-supervised methods in general. The specific lens we choose is that of parameter identifiability, which focuses on the data generative model itself and does not require extra assumptions on downstream tasks. We hope that the simple probabilistic setup in our work can serve as a starting point for theoretical inquiries, which can be extended to inform practice as modeling assumptions get relaxed.
>
> In particular, a potential pathway for gaining practical impact is to find plausible structural models for practical time-series data (e.g., natural languages, or continuous signals such as speech and EEG) which can also be theoretically analyzed, as also suggested by the reviewer. This program bears resemblance to the program of method-of-moments and sufficient statistics. An example for natural languages is Zhang & Hashimoto 21 where Gaussian graphical models are used to represent linguistic structures (such as dependencies), though we note that the question of finding proper structural models is still open for unsupervised learning at large.
>
> Reference: Zhang & Hashimoto 21: On the Inductive Bias of Masked Language Modeling: From Statistical to Syntactic Dependencies.

---

> > ### Author Response · Authors · 2022-08-08
> > **Thank you again for the review!**
> >
> > Thank you very much again for the constructive feedback! Please let us know if our response has addressed your concerns, and we are happy to discuss more if there are further questions.

---

### Official Review · Reviewer_gc3R · 2022-07-13

**Rating:** 7
**Confidence:** 4
**Soundness:** 4 excellent
**Presentation:** 4 excellent
**Contribution:** 4 excellent

**Summary:**

The paper studies "masked prediction" tasks where some tokens in a sequence are predicted from another token, akin to the popular masked language models. The authors study identifiability under two data-generating processes: an HMM with discrete states and observations, and a "G-HMM" with discrete states and continuous observations with isotropic Gaussian emissions.
They show that pairwise predictions (e.g. x_2 | x_1) do not suffice to identify parameters in the discrete HMM, while they do suffice in the G-HMM. In the discrete case, they then consider predicting 2 tokens from another token, and show that identifiability is possible in this case, by using results from identifiability of certain tensor decompositions (Kruskal, 1977).

**Questions:**

A few questions:

- it is stated that changing the loss wouldn't change the nonidentifiability result. What is the role of the loss for the identifiability results? e.g. would the softmax/cross-entropy loss also work for the discrete case?

- a related three-token setting could be to predict one token from two other tokens. Would this also provide identifiability? More generally, can anything be said about generic masking, e.g. masking 40% of a sentence? It would be helpful to comment on what could make this useful, e.g. a richer latent structure than HMMs.

- what would break down if you drop the k <= d constraint?

- regarding links to ICA in the related work, this more recent [paper](https://arxiv.org/abs/2006.12107) also seems relevant for the HMM model.

minor typos:
- L160/161: $r \to R$
- L258 increase -> increasing?
- L289: end sentence before "the second and third"?

**Limitations:**

see weaknesses

**Strengths And Weaknesses:**

Given the increasing success of self-supervised pre-training tasks, and particularly of those based on masked prediction, this work is of significant and timely interest to the community, and presents novel results on identifiability of such masked prediction tasks in discrete and Gaussian HMMs, an interesting step towards understanding such tasks. The paper is also well written and provides thorough comparisons to related literature.

Extensions to richer generative models beyond HMMs, or without the k <= d constraint, could make the work more impactful.

---

> ### Author Response · Authors · 2022-07-30
> **Thank you for your encouraging review and for the questions**
>
> We thank the reviewer for the encouraging feedback and interesting questions! We are glad you find the paper “of significant and timely interest to the community” and also that it is “well written and provides thorough comparisons to related literature.” We provide some discussions below.
> * _The effect of loss_: The choice of the loss affects the form of the optimal predictor. For instance, minimizing the squared loss matches the conditional mean (of the predicted tokens given the observed token), whereas minimizing the cross-entropy loss matches the conditional distribution. These two are equivalent for the discrete case, so the same nonidentifiability / identifiability results will hold.
> * _Other masking schemes_: This is an interesting question. For the setups we consider, we conjecture that predicting one token given two others, and some generic masks would also suffice for identifiability—though this doesn’t immediately follow from our results and certainly would be interesting to prove! We imagine that more difficult masking tasks might be required for more complex latent models. In general, we hope that our work can serve as a set of first-cut results that could benefit such future research.
> * _The $k <= d$ constraint_: this constraint is needed both for ensuring the uniqueness of tensor decompositions (i.e. being able to apply Kruskal’s theorem), and for recovering the parameters from the tensor components. For Kruskal’s theorem, we can relax this constraint as long as the sum of the Kruskal rank of the tensor components is sufficiently large (please refer to Proposition 1 for details). However, we will not be able to recover the $k \times k$ transition matrix $T$ from the tensor components, since $T$ is now under constrained (e.g. we wouldn’t be able to recover $T$ in line 295).
>
> Thank you also for pointing out the missing related work (now on line 350, highlighted in red) and the typos, we have updated the paper accordingly in the revision.

---

> > ### Author Response · Authors · 2022-08-08
> > **Thank you again for the review!**
> >
> > Thank you very much again for the insightful comments! Please let us know if our response has addressed your concerns, and we are happy to discuss more if there are further questions.

---

### Meta-Review · Area_Chair_DbVh · 2022-08-25

**Recommendation:** Accept
**Confidence:** Certain

**Metareview:**

This work investigates a theoretical way to analyze self-supervised learning. The identifiability of latent stochastic generative parameters in the context of masking self-supervised prediction tasks is investigated, where HMM and Gaussian HMM models are assumed for latent space sequences. The 3 main claims are proved by theorems. The paper does not provide experimental results in the main paper on how well the findings hold in practice, but experimental simulations are provided in the appendix.

I suggest accepting the paper, as it deals with an important topic in understanding SSL and it is of significant and timely interest to the community by presenting novel results on the identifiability of masked prediction tasks in discrete and Gaussian HMMs. Although the setup is a little narrow and strongly simplified, the paper is original enough. Further, the paper is well written and provides thorough comparisons to related literature. In future work, addressing practical impact would be a nice addition.


**Award:**

No

---

### Decision · Program_Chairs · 2022-09-14

Accept